# The Mn-motif protein MAP6d1 assembles ciliary doublet microtubules

Dharshini Gopal [1,3], Juliette Wu [1,3], Julie Delaroche[1], Christophe Bosc [1], Manon De Andrade[1], Eric Denarier [1], Gregory Effantin [2], Annie Andrieux[1], Sylvie Gory-Fauré [1] ✉, Laurence Serre [1,4] ✉ & Isabelle Arnal [1,4] ✉

Most eukaryotic cells have cilia that serve vital functions in sensing, signalling, motility. The core architecture of cilia is an array of microtubule doublets, which consist of a complete A-tubule and an incomplete B-tubule. How these structures assemble remains poorly understood. Using total internal reflection fluorescence microscopy and cryo-electron tomography, we investigate the role of MAP6d1, a brain-specific protein containing microtubule lumen-targeting Mn-motifs. We show that MAP6d1 assembles stable microtubule doublets by recruiting tubulin dimers onto the A-tubule lattice to initiate B-tubule nucleation. MAP6d1 also promotes the formation of luminal proto-filaments in singlet and doublet microtubules, a previously undescribed phenomenon that likely enhances microtubule stability. In neurons, MAP6d1 localises to the proximal part of primary cilia via its Mn-motif, with its loss resulting in shortened cilia, a characteristic of ciliopathies. MAP6d1 is thus a neuronal Mn-motif protein with a specific role in assembling microtubule doublets and regulating ciliary length.

Microtubules are one of the major classes of cytoskeletal proteins that serve a wide range of cellular functions, providing tracks for cargo transport and structural support to maintain cellular architecture and enable motility[1,2]. In order for cells to quickly reorganise their cytoskeleton in response to various triggers, dynamic microtubules alternate between phases of growth and shrinkage, known as dynamic instability[3]. In contrast, long-lived stable microtubules with reduced tubulin turnover predominate in terminally differentiated cells such as neurons, where they maintain cell morphology and support long-distance intracellular transport[4]. Particularly stable microtubules are also found in the axonemes of cilia and flagella, where they form complex arrays of doublets, ensuring the integrity and functions of these cellular appendages[5]. Motile cilia and flagella are responsible for moving cells (e.g. sperm) or a part of their environment (e.g. mucus along the trachea), while non-motile or primary cilia function as an antenna for sensory perception and

signal transduction on most vertebrate cells[6,7]. Defects in cilia assembly and structure can result in a number of diseases, collectively referred to as ciliopathies[8,9].

Cilia share a core structure, the axoneme, composed of nine doublet microtubules arranged radially around two central singlet microtubules in motile cilia, or without a central pair in primary cilia. Each doublet microtubule typically comprises one complete 13 protofilament A-tubule and one incomplete 10 protofilament B-tubule. Studies of cilia and flagella in unicellular organisms such as *Chlamydomonas* and *Tetrahymena* revealed the presence of proteins located in the microtubule lumen known as Microtubule Inner Proteins (MIPs), which have since emerged as key contributors to the remarkable stability of axonemal doublet microtubules[10–14]. In recent years, more than 60 axonemal MIPs have been identified across species, and they exhibit a variety of luminal microtubule-binding modes[15–17]. To date, however, the function of most MIPs remains obscure. More generally,

[1]Grenoble Institut Neurosciences (GIN), INSERM, U1216, CNRS, CEA, Université Grenoble Alpes, Grenoble, France. [2]Institut de Biologie Structurale (IBS), CNRS, CEA, Université Grenoble Alpes, Grenoble, France. [3]These authors contributed equally: Dharshini Gopal, Juliette Wu. [4]These authors jointly supervised this work: Laurence Serre, Isabelle Arnal. ✉e-mail: sylvie.gory-faure@univ-grenoble-alpes.fr; laurence.serre@univ-grenoble-alpes.fr; isabelle.arnal@univ-grenoble-alpes.fr

the mechanisms by which microtubule doublets are assembled and the role of MIPs in this process are still poorly understood.

MIPs are also present in neurons, where they have long been observed as intraluminal particles in both cytoplasmic and ciliary microtubules[18–23]. By analogy with axonemal MIPs, they are thought to contribute to neuronal microtubule stability, although their identity remains largely unknown. MAP6, a microtubule-stabilising factor linked to psychiatric disorders[24,25], was the first neuronal MIP located in the microtubule lumen. Intraluminal MAP6 generates highly stable microtubules that grow in a helicoidal pattern[26]. MAP6 belongs to the SAXO (Stabiliser of AXOnemal microtubules) family of proteins that contain a small helical motif in their microtubule-binding domains known as the Mn-motif[17,27–29]. This structural element has recently been recognised as a universal microtubule luminal binding feature in axonemal MIPs, highlighting its potential significance in stabilising both axonemal and neuronal microtubules. Interestingly, only two SAXO family members, MAP6 and MAP6d1 (MAP6 domain containing 1, formerly known as SL21 for 21-kDa STOP-like protein), are currently known to be expressed in the brain.

Here we focus on MAP6d1, the sole SAXO family member known to be expressed exclusively in the postnatal brain[30]. MAP6d1 contains two Mn-motifs and can protect microtubules from drug- and cold-induced depolymerisation via its Mn-motif-containing microtubule-binding domains[31]. Beyond its cellular localisation to microtubules, mitochondria, Golgi and membranes following palmitoylation[30], little is known about MAP6d1 functions. To investigate the effects of MAP6d1 on microtubules and related cellular roles, we used a combination of total internal reflection fluorescence (TIRF) microscopy-based in vitro reconstitution assays, cryo-electron tomography (cryo-ET), and cultured hippocampal neurons. Our findings uncover a direct role for MAP6d1 in the assembly and stabilisation of doublet microtubules, shedding light on its involvement in maintaining the architecture of neuronal primary cilia.

## Results

### MAP6d1 induces microtubule pausing

To gain insight into the mechanism underlying MAP6d1's ability to stabilise microtubules, we used TIRF microscopy and in vitro reconstitution assays to study microtubule dynamics (Fig. 1a and Supplementary Movie 1). We found that MAP6d1 did not affect the frequency of catastrophes but did reduce both the growth and shrinkage rates while promoting rescues, at both plus and minus ends (Fig. 1b and Supplementary Fig. 1a). The net effect was the occurrence of pauses during which the microtubules neither grew nor shrank. Pauses became more frequent at MAP6d1 concentrations of 20 nM and lasted significantly longer at concentrations of 50 nM (Fig. 1c). The longest pauses, lasting up to 20 min, blocked microtubule dynamics, with a mean paused microtubule length of about $8 \pm 4.2\ \mu m$ ($n = 45$ microtubules).

To evaluate the stability of these MAP6d1-paused microtubules, we performed dilution-induced depolymerisation assays. Microtubules assembled with tubulin alone quickly depolymerised within seconds upon buffer perfusion (Fig. 1d, left), whereas those assembled in the presence of MAP6d1 resisted depolymerisation with three distinct behaviours (Fig. 1d, right). One population of microtubules (37.4%) remained in a paused state for the entire duration of the movie (30 min) following buffer perfusion; a second (29.3%) showed pauses interrupted by small catastrophes; and a third population (32.1%) continuously but slowly depolymerised after buffer perfusion (Fig. 1e). The shrinkage rate of these slowly depolymerising microtubules was 50-fold slower than that observed for microtubules assembled without MAP6d1 (Fig. 1f).

We next performed a two-step perfusion TIRF experiment, polymerising microtubules from red tubulin in the presence of MAP6d1, then exchanging the solution with green-fluorescent tubulin before recording. In this experiment, approximately 70% of the dynamic green-fluorescent microtubules, polymerising from the extremities of

MAP6d1 co-assembled microtubules, depolymerised up to the junction with the MAP6d1-assembled microtubule lattice, which served as a rescue point (Fig. 1g). Thus, microtubules assembled in the presence of MAP6d1 resist buffer-induced depolymerisation and are stable enough to stop the depolymerisation of a dynamic microtubule.

Overall, these results reveal that MAP6d1 is a strong microtubule stabiliser.

### MAP6d1 induces the assembly of doublet microtubules by recruiting tubulin

To investigate how MAP6d1 influences microtubule architecture, we performed negative staining electron microscopy on microtubules polymerised in the presence or absence of MAP6d1 (Fig. 2a). To our surprise, MAP6d1 enabled the formation of doublet microtubules, which were not seen with microtubules polymerised with tubulin alone. Cryo-ET revealed distinct microtubule architectures, including doublet microtubules with varying B-tubules, as shown in tomograms in (Fig. 2b and Supplementary Fig. 2). The complete microtubules, i.e. the A-tubules of doublet microtubules in our dataset, were consistently composed of 14 protofilaments, whereas the z-slices of the example tomograms delineated eleven classes of B-tubule with different numbers of protofilaments. Most B-tubules had 7–11 protofilaments, with 10 protofilaments being the most abundant class (~40 % of the extracted particles). This heterogeneity in B-tubule architecture could represent different stages of doublet microtubule assembly. Notably, the B-tubule was highly flexible, as its extremity did not anchor back to the A-tubule but exhibited global curvature similar to that observed in vivo (13.15 nm versus 13.50 nm, respectively, Fig. 2c and see below). Using subtomogram averaging, we reconstructed a doublet microtubule from these data, showing an A-tubule with 14 protofilaments and a B-tubule with 10 protofilaments (Fig. 2d).

To better understand the underlying mechanism of doublet microtubule formation, we performed tubulin recruitment assays using TIRF microscopy (Fig. 2e). Red-fluorescent pre-polymerised microtubules were incubated with free green-fluorescent tubulin. In the absence of MAP6d1, there was no tubulin recruitment (Fig. 2e, *left*), but with MAP6d1 we observed recruitment of free tubulin onto 57% of the pre-polymerised microtubules (Fig. 2e, f). We also noticed elongation of this recruited tubulin along the pre-existing microtubules (Supplementary Movie 2). To investigate the impact of tubulin recruitment on microtubule structure, we next imaged these samples by negative staining electron microscopy. In the presence of MAP6d1, 40% of microtubules were doublets, but without MAP6d1, there appeared only singlet microtubules (Fig. 2e, f). This confirms that the ability of MAP6d1 to assemble doublet microtubules depends on its capacity to recruit tubulin dimers along the microtubule lattice.

### The Mn-motif of MAP6d1 determines its microtubule-regulating activities

We analysed the domains previously identified as essential for properties of MAP6d1 (Supplementary Fig. 3). The mutant MAP6d1-ΔMn2, which deletes the microtubule-binding domain encompassing the second Mn-motif and its C-terminal flanking region, conserved in MAP6[30], abolished the microtubule-stabilising activity of MAP6d1 (Fig. 3a, b and Supplementary Fig. 1b). This deletion also eliminated the ability of MAP6d1 to recruit tubulin dimers along the microtubule lattice and to promote microtubule doublet assembly (Fig. 3c, d). To assess whether these effects were directly linked to the Mn-motif, we designed a mutant in which the seven residues of the second Mn-motif were replaced by alanine (MAP6d1-Mn2-7A). This mutant was also unable to stabilise microtubules, promote doublet assembly or recruit tubulin onto the microtubule lattice (Fig. 3). We then assessed the role of the first Mn-motif using a similar alanine substitution mutant (MAP6d1-Mn1-7A). This mutant retained residual microtubule-stabilising activity, as evidenced by low growth and shrinkage rates

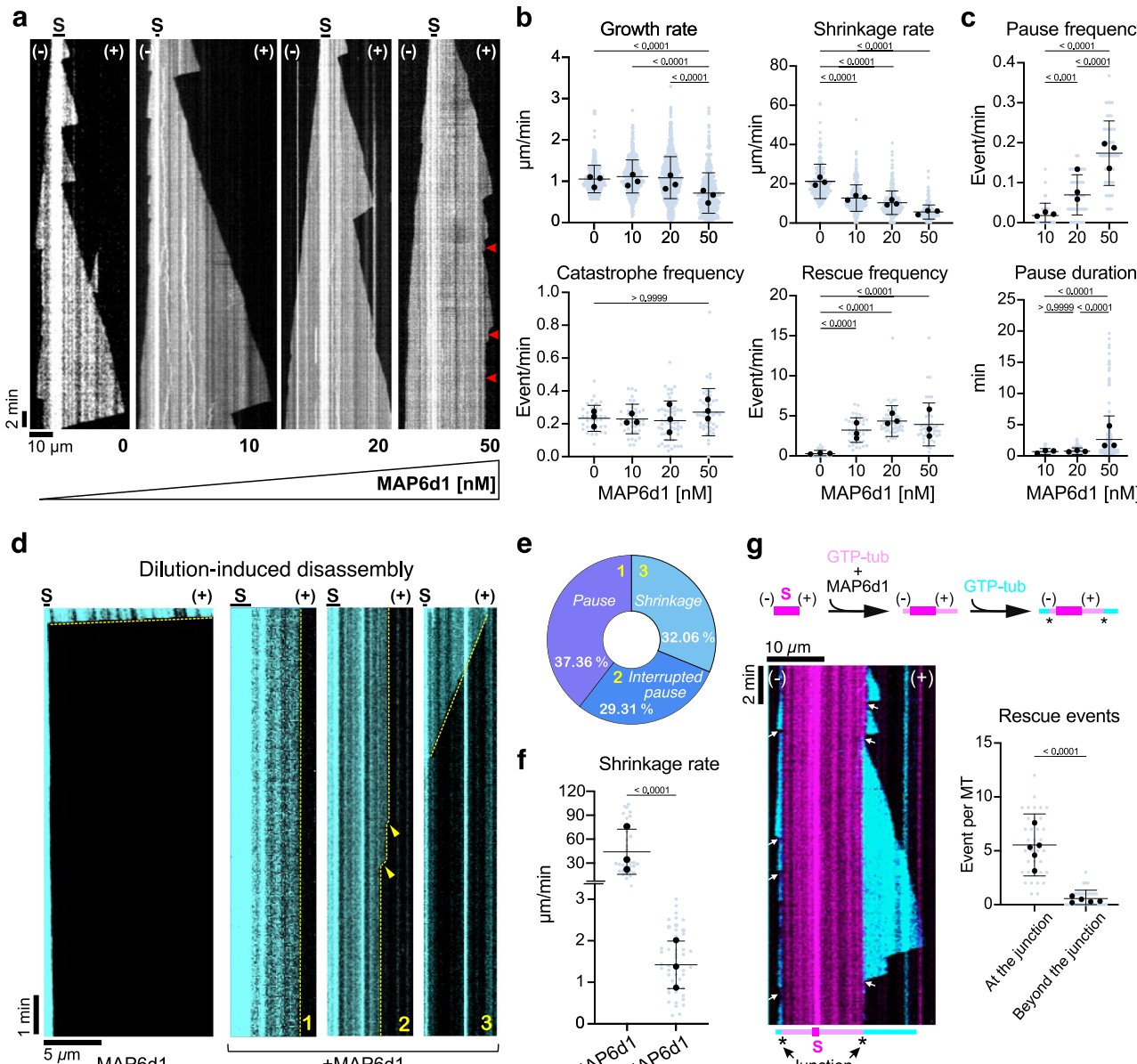

**Fig. 1 | MAP6d1 induces microtubule pauses. a** Representative kymographs of microtubules grown from seeds (S) and 12 μM tubulin with increasing MAP6d1 concentrations. Arrowheads indicate pauses. **b** Graphs showing the growth rate (*n* = 301, 427, 712, and 364 growth events for 0, 10, 20 and 50 nM MAP6d1, respectively), shrinkage rate (*n* = 168, 274, 334, and 264 shrinkage events), catastrophe and rescue frequencies (*n* = 36, 41, 54 and 45 frequencies) of microtubules assembled with or without MAP6d1. **c** Graphs showing the pause frequency (*n* = 41, 54, and 45 pause frequencies, for 10, 20, and 50 nM MAP6d1, respectively) and durations (*n* = 24, 121, and 245 pause events) in similar conditions. In the absence of MAP6d1, microtubules did not exhibit any pauses. **d** Representative kymographs of microtubules undergoing disassembly after tubulin dilution. Microtubules were grown from seeds (S) and 25 μM tubulin (control, *left*) or 12 μM tubulin and 50 nM MAP6d1 (*right*). After dilution, three populations were observed in the presence of MAP6d1: continuous pause after dilution, pauses interrupted by small catastrophes (arrowheads), and continuous slow depolymerisation. **e** Percentages of the three populations presented in (**d**). **f** Shrinkage rate of control and population 3 microtubules shown in (**d**) (*n* = 32 and 50 shrinkage events, respectively). **g** Kymograph showing rescue events of a microtubule grown with 12 μM GTP-tubulin (cyan) at the extremity of a microtubule assembled with 12 μM GTP-tubulin and 50 nM MAP6d1 (magenta). The experimental procedure is depicted above the kymograph. The kymograph shows the behaviour of a dynamic microtubule following GTP-tubulin perfusion. Arrows indicate rescues occurring at the junction (*) between the MAP6d1-assembled microtubule (magenta) and the newly growing dynamic microtubule (cyan). The graph represents the number of rescues occurring at or beyond the junction (*n* = 194 and 20, respectively). For graphs in (**b**), (**c**), (**f**), (**g**), bars represent mean ± SD from at least three independent experiments. Circles represent the mean of each individual experiment. *p* values are indicated (Kruskal-Wallis ANOVA followed by post hoc Dunn's multiple comparison test). Source data are provided as a Source Data file.

and a higher binding affinity for microtubules (Supplementary Fig. 4a–d); it did not, however, promote pause events, recruit tubulin to the microtubule lattice or assemble doublet microtubules (Supplementary Fig. 4e, f). Overall, these results indicate that both Mn motifs contribute to the observed microtubule-regulatory roles of MAP6d1, with the Mn2 motif playing a dominant role.

## The N-terminal domain of MAP6d1 is required for microtubule pausing and doublet formation

Next, we compared the microtubule-regulatory activity of MAP6d1 to that of a mutant lacking its N-terminal domain (MAP6d1-Δ2-35), which is conserved in MAP6 and involved in MAP6-mediated microtubule stabilisation and luminal particle formation[26,30]. Relative to the control

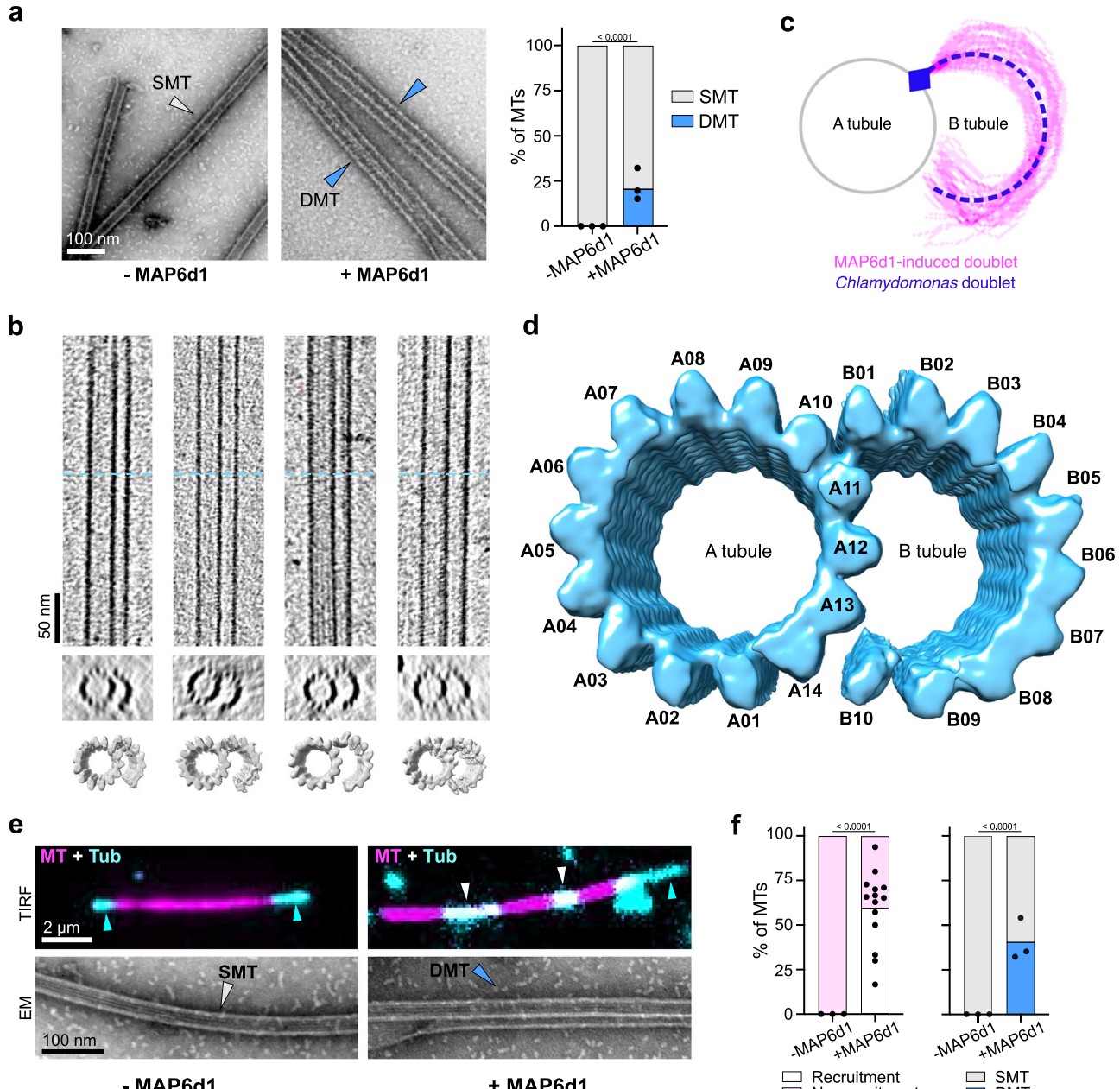

**Fig. 2 | MAP6d1 induces the assembly of doublet microtubules by recruiting tubulin. a** Negative staining electron microscopy images of microtubules assembled with 25 µM tubulin in absence or presence of 250 nM MAP6d1. White and blue arrowheads indicate singlet (SMT) and doublet (DMT) microtubules, respectively. The graph represents the percentage of SMT and DMT assembled without and with MAP6d1 (total measured lengths of 304 µm and 401 µm, respectively, from 3 experiments). **b** Examples of cryo-electron tomograms of doublet microtubules assembled with MAP6d1. The top panel shows longitudinal sections of tomograms along with a transverse section at the position of the blue dashed lines. A subtomogram average map is presented below each tomogram. **c** B-tubule traces of MAP6d1-induced doublet microtubules starting at the outer junction (blue square) and overlapping the curvature of the B-tubule of the *Chlamydomonas* flagellar doublet microtubules (EMD-40621) for comparison (*n* = 45 B-tubule curvatures). **d** Reconstruction of MAP6d1-induced doublet microtubules obtained by combining the maps of the doublet microtubule and the B-tubule. **e** *Top*: snapshots of TIRF microscopy recorded movies of 0.325 µM tubulin (cyan) mixed with GMPCPP-stabilised microtubule seeds (magenta) in the absence or presence of 100 nM MAP6d1 and 50 µM GMPCPP. White and blue arrowheads indicate tubulin recruitment and polymerisation at microtubule ends, respectively. *Bottom*: negative staining electron microscopy images of microtubules assembled in the same conditions as for TIRF microscopy. White and blue arrowheads indicate SMT and DMT, respectively. MT microtubule, Tub tubulin. **f** *Left*: percentage of microtubules that have or have not recruited tubulin (*n* = 57 and 282 microtubules from 3 and 14 independent experiments, respectively). Black dots represent the percentage of microtubules with recruitment for each experiment. *Right*: percentage of singlet and doublet microtubules in the control and with MAP6d1. The total measured lengths of microtubules were 225 µm and 543 µm for the control and with MAP6d1, respectively (three independent experiments). Black dots represent the percentage of doublet microtubules for each experiment. In (**a**, **f**), Fischer's exact contingency test (two-sided) was applied (*p* values are indicated). Source data are provided as a Source Data file.

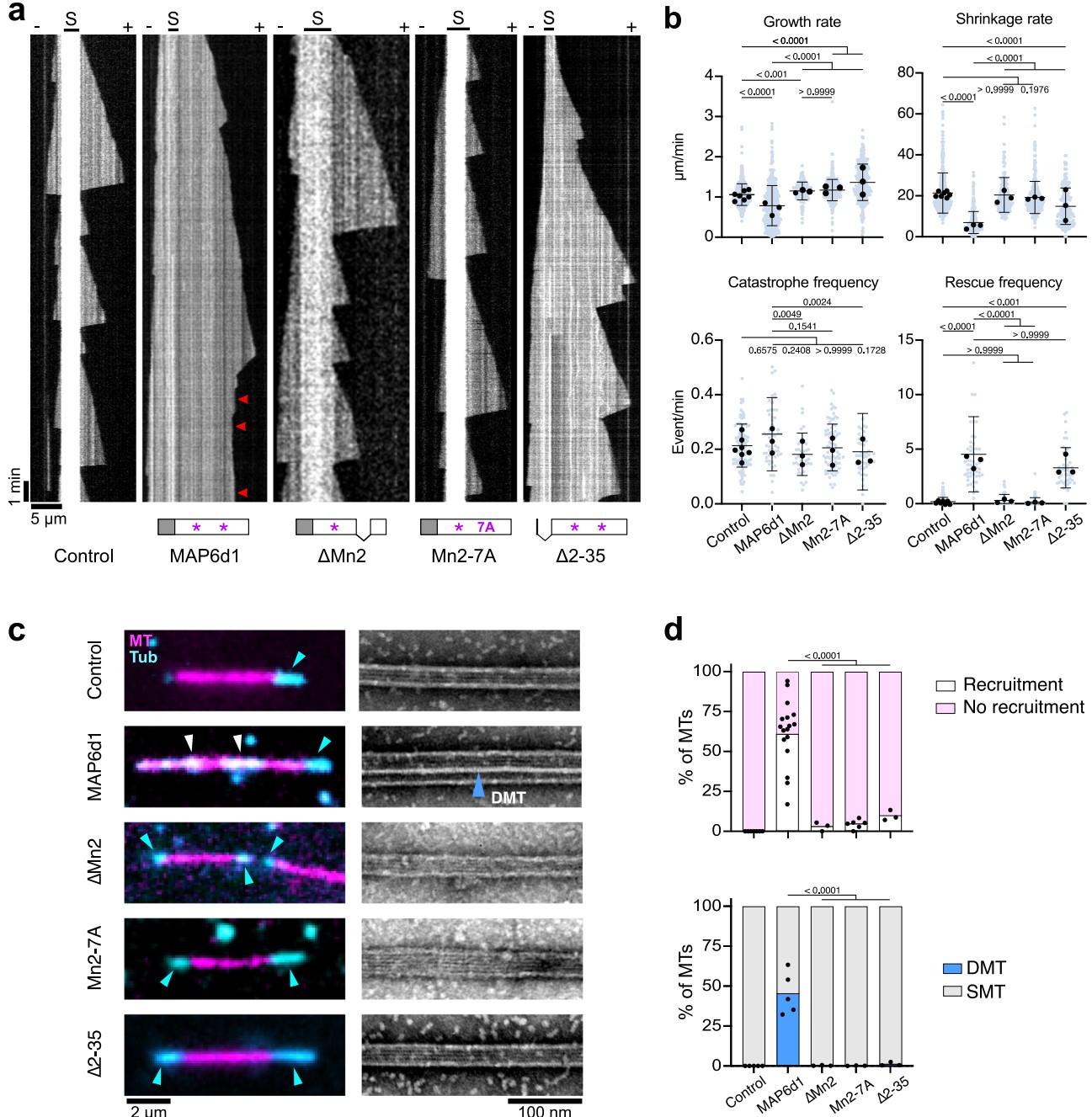

**Fig. 3 | The microtubule-regulating activities of MAP6d1 depend on its Mn-motif and N-terminus. a** Kymographs depicting microtubules grown from seeds (S) and 12 µM tubulin alone or with 50 nM MAP6d1, 300 nM MAP6d1-ΔMn2, 300 nM MAP6d1-Mn2-7A or 50 nM MAP6d1-Δ2-35 (n = 105, 57, 81, 49 and 38 microtubules, respectively). Arrowheads indicate microtubule pauses. **b** Dynamical parameters of microtubules assembled without or with MAP6d1, MAP6d1-ΔMn2 MAP6d1-Mn2-7A and MAP6d1-Δ2-35: growth rate (n = 556, 437, 190, 397 and 286 growth events, respectively), shrinkage rate (n = 444, 284, 162, 348 and 197 shrinkage events, respectively), catastrophe and rescue frequencies (n = 105, 56, 38, 81 and 48 frequencies, respectively). Bars represent mean ± SD from at least three independent experiments. Circles represent the mean of each individual experiment. *p* values are indicated (Kruskal-Wallis ANOVA followed by post hoc Dunn's multiple comparison test). **c** *Left*: snapshots of TIRF movies of 0.325 µM tubulin (cyan) mixed with GMPCPP seeds (magenta) in the absence or presence of 100 nM MAP6d1, 300 nM MAP6d1-ΔMn2 300 nM MAP6d1-Mn2-7A and 100 nM MAP6d1-Δ2-35. White and blue arrowheads indicate tubulin

recruitment and polymerisation at the seed extremities, respectively. *Right*: Negative staining electron microscopy images of microtubules assembled in the same conditions as for TIRF microscopy. The blue arrowhead indicates a doublet microtubule (DMT). MT microtubule, Tub tubulin. **d** *Top*: percentage of microtubules with or without recruited tubulin (n = 57, 282, 63, 164 and 152 microtubules from 7, 17, 3, 6 and 3 independent experiments for the control and with MAP6d1, MAP6d1-ΔMn2, MAP6d1-Mn2-7A and MAP6d1-Δ2-35, respectively). Black dots represent the percentage of microtubules with recruitment for each experiment. *Bottom*: percentage of singlet (SMT) and doublet (DMT) microtubules in the control and with MAP6d1 and its mutants (total measured lengths of 361, 752, 149, 221 and 198 µm for MAP6d1, MAP6d1-ΔMn2, MAP6d1-Mn2-7A and MAP6d1-Δ2-35, respectively). Black dots represent the percentage of doublet microtubules for each experiment. Fischer's exact contingency test (two-sided) was applied. All statistical analyses were performed from at least three independent experiments and *p* values are indicated. Source data are provided as a Source Data file.

condition, MAP6d1-Δ2-35 stabilised microtubules similarly to MAP6d1 by increasing the rescue frequency and reducing the shrinkage rate (Fig. 3a, b and Supplementary Fig. 1b) while also promoting the growth rate, leading to persistent microtubule growth. Unlike MAP6d1, however, MAP6d1-Δ2-35 did not induce microtubule pausing (Fig. 3a), and it suppressed doublet microtubule formation (only ~1% vs. the ~45% with MAP6d1; Fig. 3c, d). Consistent with these observations, this mutant weakly recruited free tubulin along stabilised microtubules (~10% versus ~61% for MAP6d1) (Fig. 3c, d). Thus, although the N-terminal domain is not needed for stabilising microtubules, it is essential to induce pauses by decreasing the microtubule growth rate and to form doublets by recruiting tubulin.

### MAP6d1 localises to microtubule doublets in neuronal primary cilia and regulates ciliary length

Building on evidence of MAP6d1's ability to assemble microtubule doublets in vitro, we moved to studies in neurons, focusing on primary cilia. We compared primary cilia from cultured hippocampal neurons derived from Wild-Type (WT) and MAP6d1-KnockOut (KO) mice (Fig. 4a, b). The percentage of ciliated neurons was similar in both cases (Supplementary Fig. 5a). At 4 days in vitro (DIV), ciliary length (4.3 ± 1.861 μm) was also similar to that observed in WT (4.4 ± 1.746 μm) (Fig. 4a). Given that MAP6d1 expression begins around 7 DIV[30], we re-examined ciliary length at 9 DIV, and found that MAP6d1-KO cilia had become 15% shorter than those in WT neurons (5.38 ± 2.022 μm to 4.7 ± 1.695 μm) (Fig. 4b); this difference persisted through 23 DIV (Supplementary Fig. 5b).

To determine the localisation of MAP6d1 in neurons, we then expressed ectopic GFP-tagged MAP6d1 in the hippocampal cells. We observed that MAP6d1-GFP localised to the proximal part of the primary cilia in all transfected cells and led to cilia ~24% longer than those in control cells (Fig. 4c and Supplementary Fig. 5c). This asymmetric distribution contrasted with that of the ciliary marker Arl13b, which spans the entire length of the cilium. The proximal localisation of MAP6d1 corresponds to the microtubule doublet region identified by cryo-ET showing that doublet microtubules lose their B-tubule, eventually leaving only singlet A-tubules in the distal part of the cilia[32,33]. Expansion microscopy showed that MAP6d1-GFP co-localised with polyglutamylated tubulin—a post-translational modification (PTM) enriched on the B-tubules (and therefore doublet microtubules) in motile cilia[34–36]— in the central region of the cilia, positioned between the two distinct bands of the membrane-associated protein Adenylate Cyclase that delineate the borders of the cilia (Fig. 4d, e).

To further investigate MAP6d1's preferential proximal enrichment, we performed immunofluorescence co-staining with PTMs of tubulin known to accumulate in ciliary axonemes. We confirmed that MAP6d1-GFP co-localised with polyglutamylated tubulin but not with tyrosinated tubulin, a PTM associated with dynamic microtubules[37,38] and enriched at the distal end of the cilia (Fig. 4f, g). This finding is consistent with previous observations that single dynamic microtubules are found predominantly at the ciliary tips[39,40]. To assess whether MAP6d1 influences the distribution of doublet or singlet MTs in cilia, we compared the distribution of tubulin PTMs associated with these microtubule architectures in wild-type and MAP6d1-KO neurons. Both polyglutamylated and tyrosinated tubulin exhibited similar patterns in the two types of neurons, despite MAP6d1-KO neurons bearing shorter cilia (Supplementary Fig. 5d).

To determine whether MAP6d1's Mn-motif or N-terminal influence ciliary localisation, we expressed GPF-tagged-ΔMn2 and Δ2-35 mutants in hippocampal neurons (Fig. 4c and Supplementary Fig. 5c). Unlike MAP6d1, MAP6d1-ΔMn2-GFP failed to localise to the cilia, which were similar in length to those in wild-type cells. In contrast, MAP6d1-Δ2-35-GFP was present all along the cilia, which were twice as long as those in control neurons. The distribution of tubulin PTMs remained similar to that observed in MAP6d1-expressing neurons (Supplementary Fig. 5e).

MAP6d1's ciliary localisation therefore requires the Mn2 motif, and its targeting of the proximal cilia, associated with doublet microtubules, is linked to the N-terminal domain. Moreover, these results suggest that MAP6d1 plays a role in regulating ciliary length in neurons, likely by assembling and stabilising microtubule doublet architecture.

### MAP6d1 assembles protofilaments in the lumen of singlet and doublet microtubules

Tomograms of microtubules assembled in the presence of MAP6d1 revealed peculiar structures within the lumen of both doublet and singlet microtubules (Fig. 5a). These structures correspond to two adjacent tubulin protofilaments that would interact with the microtubule internal surface. Indeed, subtomogram averages applied to these MAP6d1-induced structural entities allowed to obtain reconstructions of doublet and singlet microtubules with two luminal protofilaments (Fig. 5b, c). These reconstructions indicate that the luminal protofilaments in the A-tubule of the doublet face protofilaments A10–A13, adjacent to the putative seam (Fig. 5b) which we positioned between protofilaments A9 and A10 in the doublet microtubule, based on previous structural studies[11,41]. We did not detect any obvious lattice defects in proximity to these luminal protofilaments (in a total of 220 microtubules), suggesting that luminal protofilaments are formed during microtubule copolymerisation with MAP6d1 rather than through diffusion via breaks in the lattice.

We next compared the B-tubule curvatures of doublet microtubules with and without luminal protofilaments (Fig. 5d) and found that the mean radii of the B-tubules with luminal protofilaments increased by ~15% compared to those without (Fig. 5d). As luminal protofilaments are localised in the A-tubule right next to where the B-tubule begins, known as the outer junction, they may influence the conformation of the B-tubule.

Because protofilaments within the microtubule lumen have not been previously described, we used cryo-electron tomography to look for the presence of such microtubule architectures in mature neurons (Fig. 5e and Supplementary Fig. 6). We observed a few rare examples of singlet microtubule structures containing filamentous densities that resemble the luminal protofilaments induced in vitro by MAP6d1, in neuritic extensions (Fig. 5f and Supplementary Fig. 6). We propose that these luminal protofilaments could help explain the strong stability of neuronal microtubules.

## Discussion

These data reveal that MAP6d1 regulates microtubule architecture within neuronal primary cilia. As MAP6d1 is expressed in the brain postnatally, it likely contributes to ciliary elongation and maturation, a process that lasts for several weeks after birth[42]. More specifically, MAP6d1 induces the assembly of doublet microtubules and is able to independently reconstitute these ciliary structures. We also observed a previously undescribed phenomenon of protofilaments within the lumen that we suspect lend extra strength to neuronal microtubules.

MAP6d1 forms doublet microtubules through tubulin recruitment on the A-tubule lattice, facilitating the nucleation of the B-tubule. Our data support a model in which MAP6d1 behaves both as a MAP, in that it binds to the surface of the microtubule lattice, and as a MIP, in that it contains two universal lumen-targeting Mn-motifs. These features, respectively, are necessary for its ability to recruit tubulin to the MT lattice and form intraluminal filaments. We propose that this 'dual binding' mode is key to MAP6d1's unique ability to assemble doublet microtubules by bridging the A-tubule to the B-tubule (Fig. 6). Given that the C-terminal tail of tubulin plays a crucial role in doublet microtubule assembly[43,44], it may be that MAP6d1 binding to the A-tubule triggers a conformational change in the tubulin C-terminal

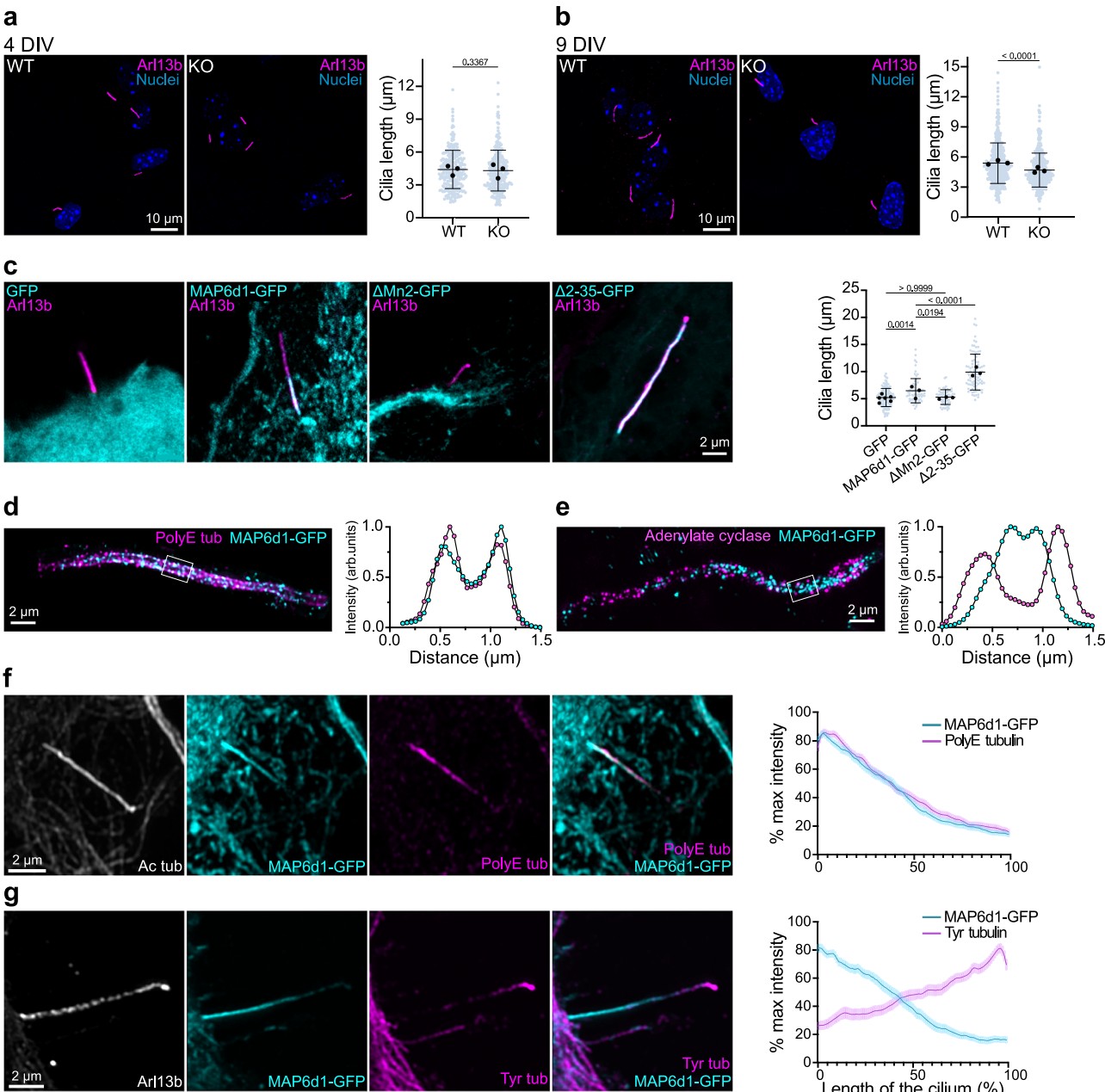

**Fig. 4 | MAP6d1 preferentially localises to microtubule doublets in neuronal primary cilia and regulates ciliary length. a** Representative images of 4 days in vitro (DIV) wild-type (WT) and MAP6d1-knockout (KO) neurons stained against cilia marker Arl13b and nuclear dye Hoechst. The graph represents cilia length of WT (*n* = 253) and KO neurons (*n* = 262). **b** Same as (**a**) for 9 DIV neurons (WT: *n* = 407, KO: *n* = 398). In (**a**, **b**), bars represent mean ± SD from three independent experiments. Circles represent individual experiment means. *p* values are indicated (Mann-Whitney's test). **c** Localisation of ectopic GFP, MAP6d1-GFP, MAP6d1-ΔMn2-GFP and MAP6d1-Δ2-35-GFP compared to ciliary marker Arl13b, in 7 DIV neurons. The graph represents cilia length in neurons expressing GFP (*n* = 128 cilia), MAP6d1-GFP (*n* = 68 cilia), MAP6d1-ΔMn2-GFP (*n* = 59 cilia), MAP6d1-Δ2-35-GFP (*n* = 86 cilia) from at least three independent experiments. The graph presents individual data points with bars representing mean ± SD. Circles represent individual experiment means. *p* values are indicated (One-way ANOVA and pairwise comparisons). **d**, **e** 4X Expansion microscopy images of MAP6d1's ciliary localisation with PolyE tubulin (**d**) or the cilia membrane marker Adenylate cyclase (**e**). Plot profiles drawn perpendicular to the cilia within the white rectangles. Images are representative of 20 cilia from two independent experiments. **f** Representative images of primary cilia from 7 DIV neurons expressing ectopic MAP6d1-GFP and stained for acetylated (Ac tub) and polyglutamylated tubulin (PolyE tub). The graph represents the percentage of normalised maximum fluorescence of MAP6d1-GFP and polyglutamylated tubulin (PolyE tub) along the ciliary length determined by acetylated tubulin (Ac tub). Values represent mean ± SEM of *n* = 58 cilia from 3 independent experiments. **g** Representative images of primary cilia from 7 DIV neurons expressing ectopic MAP6d1-GFP stained for Arl13b and tyrosinated tubulin (Tyr tub). The graph represents the percentage of normalised maximum fluorescence of MAP6d1-GFP and tyrosinated tubulin along the ciliary length determined by Arl13b. Bars represent mean ± SEM of *n* = 46 cilia from 3 experiments. Source data are provided as a Source Data file.

tail, promoting tubulin recruitment and subsequent B-tubule nucleation. Although removal of the C-terminal tail of tubulin allows B-tubule nucleation from multiple sites on the A-tubule[43], MAP6d1 binding is sufficient to restrict B-tubule nucleation to a single A-tubule protofilament. MAP6d1 therefore appears to provide positional information to recruit tubulin at a specific site on the A-tubule lattice, likely related to its binding near the seam region. Since deletion of the N-terminal part of MAP6d1 prevents both doublet formation and tubulin recruitment, we suspect the N-terminal participates in this local C-terminal tubulin conformational change.

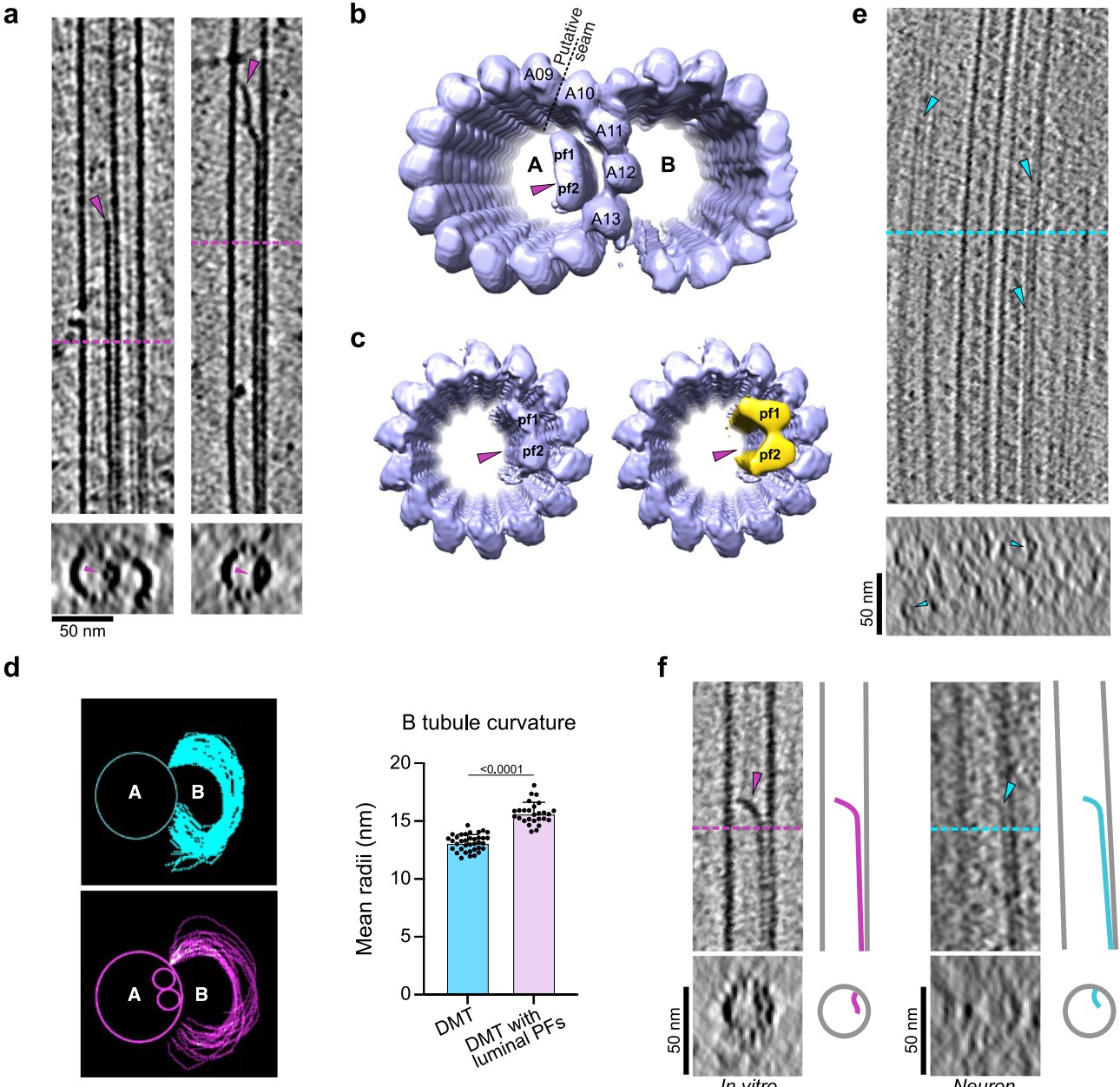

**Fig. 5 | MAP6d1 assembles protofilaments in the microtubule lumen.**
**a** Examples of tomograms of doublet (left) and singlet (right) microtubules containing luminal protofilaments. **b** Subtomogram averaged model of doublet microtubules containing luminal protofilaments (pf). The putative location of the microtubule seam is indicated by a dashed line. **c** Subtomogram averaged model of singlet microtubules containing luminal protofilament (left) and its superposition with the focused map of the luminal protofilaments (yellow, right). **d** B-tubule curvatures traced for doublet microtubules without (*top*) and with (*bottom*) luminal protofilaments. The graph shows the mean radii of the B-tubule curvatures ($n = 36$ and 27 for doublet microtubules without and with luminal protofilaments, respectively). Bars represent mean ± SD. Mann-Whitney test was applied (*p* values are indicated). **e** Example of a tomogram showing neuronal microtubules containing luminal protofilaments (blue arrowheads). **f** Comparison of microtubules containing luminal protofilaments assembled with MAP6d1 in vitro (*left*) with those observed in neurons (*right*) and their corresponding schemes. Pink and blue arrowheads show luminal protofilaments in microtubules assembled in vitro and in neurons, respectively. Source data are provided as a Source Data file.

MAP6d1 has one particular feature that may help organise both singlet and doublet microtubules in neurons: it is one of the few proteins, along with the neuronal kinesin-4 (KIF21B) and the ciliary MIP CSPP1, that can induce microtubule pausing[45,46]. It is worth noting in this context that deletion of CSPP1 in zebrafish reduces the length of primary cilia[47], similar to what we observed in MAP6d1-KO neurons. On the other hand, the MAP6d1-Δ2-35 mutant failed to induce pausing in vitro and doubled the length of primary cilia when overexpressed in neurons. It thus seems that microtubule pauses are important for maintaining

optimal ciliary length, which varies considerably in response to physiological and pathological changes[48]. Understanding the mechanism by which MAP6d1 induces pauses will require further study, but it could result from a combination of a strong lattice-stabilising effect that prevents tubulin release, as indicated by MAP6d1's inhibition of microtubule shrinkage, and a modified tip conformation that hinders tubulin incorporation, as proposed for KIF21B[46].

MAP6d1 may also contribute to microtubule stability in a unique way by assembling protofilaments within the microtubule lumen. In

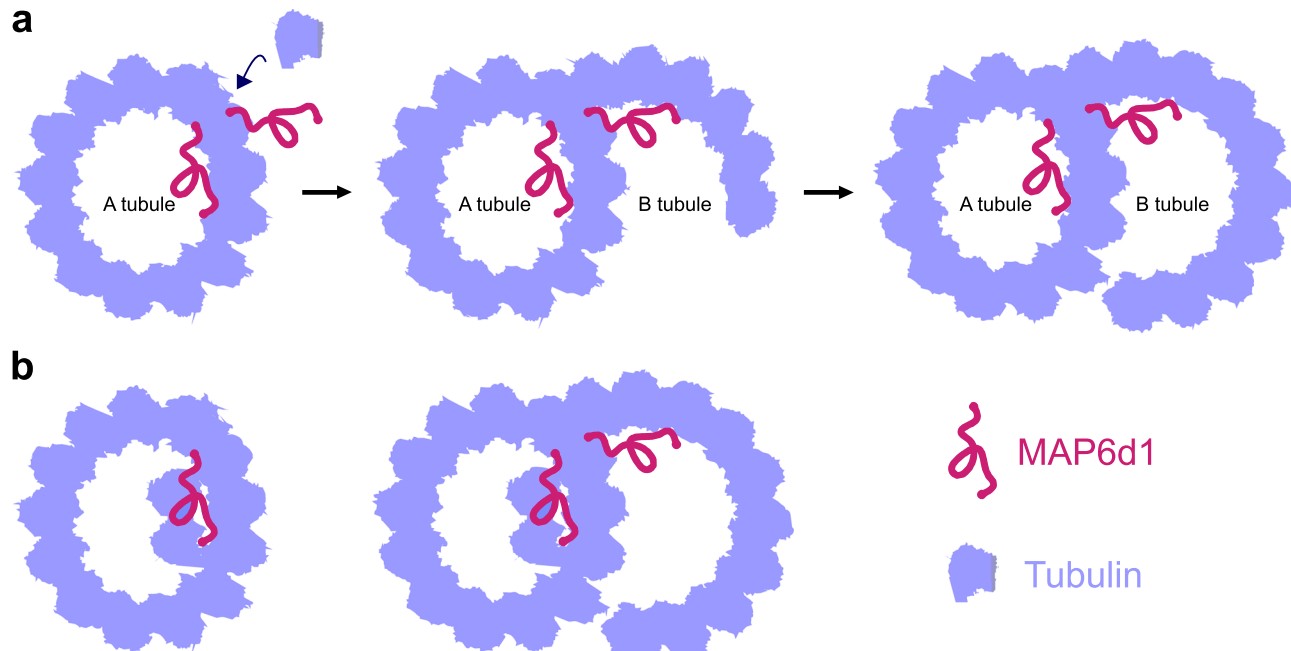

**Fig. 6 | Models of MAP6d1-microtubule interactions leading to the formation of the B-tubule and luminal protofilaments. a** MAP6d1 binds to the A-tubule lattice, recruits free tubulin dimers to initiate the B-tubule and forms a doublet microtubule. **b** During co-polymerisation with microtubules, MAP6d1 assembles protofilaments in the lumen.

doublet microtubules, luminal protofilaments in the A-tubule always localise near the outer junction, reinforcing the positional specificity of MAP6d1 binding at the B-tubule nucleation site, as discussed above. Such luminal protofilaments could constrain the architecture of the B-tubule or stabilise the doublet outer junction, similar to the organisation of MIPs in axonemal doublets[10,17,49].

MAP6d1 is part of the SAXO family, which is defined by the universal microtubule lumen-targeting Mn-motif. Most proteins in this family are found within the doublet microtubules of cilia and flagella, where they are thought to stabilise these structures[11,16,17,31,49]. Although structural studies have mapped the distribution of these proteins in doublets, the mechanisms by which they contribute to microtubule stabilisation and/or doublet formation remain largely unexplored. MAP6d1 stands out as a Mn-motif protein with a specific role in primary cilia and more generally, in the assembly of microtubule doublets. Together, these findings mark an important step toward uncovering the molecular mechanisms driving microtubule doublet formation and understanding the specialised functions of Mn-motif proteins across different cell types.

## Methods
### Ethical statement
All experiments involving animals were approved by the local animal welfare committee (Comité Local CEEA n°44 – APAFIS number 45896-2023111317017897) and in compliance with the European Community Council Directive (directive 2010/63/EU).

### Plasmid constructs
cDNAs encoding mouse MAP6d1 (full length, aa 1-191), MAP6d1-Δ2-35 (deletion of residues 2–35), MAP6d1-ΔMn2 (deletion of residues 123 to 145) and MAP6d1-Mn2-7A (amino acids 123 to 128 replaced by alanine) were amplified by PCR using Phusion High-Fidelity DNA polymerase (Thermo Scientific) and cloned into pET42a vector with In-Fusion HD cloning kit (Clontech). Primer sequences are listed in Supplementary Table 1. All encoded proteins exhibited an 8-histidine tag at their C-terminal end. For eukaryotic expression, we used plasmids of mouse MAP6d1-GFP, MAP6d1-Δ2-35-GFP and MAP6d1-ΔMn2-GFP[30].

### Tubulin preparation
Tubulin was purified from bovine brain and coupled with biotin, ATTO-565 or ATTO-496 as previously described[50,51]. Briefly, brain tissue was homogenised in cold AB buffer (0.1 M Pipes, 0.5 mM $MgCl_2$, 2 mM EGTA, 0.1 mM EDTA, pH 6.8; 1 mL per gram of tissue) and centrifuged at $100,000 \times g$ for 90 min at 4 °C. Tubulin in the supernatant was polymerised by adding 33% (v/v) glycerol, 0.2 mM GTP, 1.5 mM ATP, and 3 mM $MgCl_2$, followed by incubation at 37 °C for 60 min. The mixture was then centrifuged at $100,000 \times g$ for 90 min at 35 °C. Pellets were resuspended in AB buffer and incubated for 30 min at 4 °C to induce microtubule depolymerisation, then centrifuged at $140,000 \times g$ for 30 min at 4 °C. The supernatant was supplemented with 33% glycerol, 1 mM GTP, 0.5 mM ATP, and 3 mM $MgCl_2$, and subjected to a second polymerisation–depolymerisation cycle. Microtubule-associated proteins (MAPs) in the solution were separated from tubulin by cation-exchange chromatography using Fractogel EMD $SO_3^-$ resin (VWR) equilibrated in BRB80 buffer (80 mM Pipes, 1 mM EGTA, 1 mM $MgCl_2$, 1 mM DTT, pH 6.8) supplemented with 0.1 mM GTP. Fractions containing tubulin were pooled, supplemented with 33% glycerol and 1 mM GTP, and incubated for 1 h at 37 °C, before been poured onto a BRB80/60% glycerol cushion and centrifuged at $100,000 \times g$ at 35 °C for 90 min. Pelleted microtubules were resuspended in cold BRB80 buffer and incubated for 30 min at 4 °C to induce complete depolymerisation. Tubulin aliquots (5–10 μL) were flash-frozen and stored in liquid nitrogen.

For tubulin labelling, sulfo-NHS-LC-biotin (Thermo Scientific) or NHS-ester-ATTO-565/-ATTO-488 (ATTO-TEC Gmbh, Germany) was dissolved in anhydrous DMSO at 100 mM and mixed (5% v/v) with microtubules polymerized from purified tubulin, in a 100 mM Na-Hepes buffer (pH 8.6) containing 40% glycerol, at 35 °C for 10 min. The labelling reaction was stopped by adding two volumes of freshly prepared 100 mM potassium glutamate solution. Labelled microtubules were layered onto a BRB80/60% glycerol cushion and

centrifuged at $100,000 \times g$ for 30 min at 35 °C. Pellets were resuspended in cold BRB80, centrifuged at $100,000 \times g$ for 15 min at 4 °C, and subjected to an additional polymerisation–depolymerisation cycle before aliquoting, flash-freezing, and storing in liquid nitrogen.

## Recombinant protein expression and purification

MAP6d1 and its mutants were expressed in BL21 (DE3) Star *Escherichia coli* cells. After induction by 0.1 mM IPTG overnight at 18 °C, cells were sonicated in a lysis buffer (50 mM Hepes pH 7.4, 200 mM KCl, 0.1% Triton, and complete EDTA-free protease inhibitor cocktail tablets [GE, Healthcare]) before centrifugation at $160,000 \times g$ for 30 min at 4 °C. The clarified lysate was incubated with a cobalt affinity resin (Talon, Clontech) and the protein was eluted by increasing concentrations of imidazole. Collected fractions were concentrated, dialysed against 50 mM Hepes pH 7.4, 200 mM KCl, 20 mM DTT, centrifuged at $100,000 \times g$ for 5 min at 4 °C before freezing in liquid nitrogen. Protein concentrations were determined by comparing band intensities on SDS-PAGE gels to a standard curve generated by loading a range of known BSA concentrations on the same gels. The intensities of the bands were quantified with ImageJ software[52] (version 2.160/1.54p). Purified proteins used in this study are shown in Supplementary Figs. 3 and 4.

## Preparation of microtubule seeds and Taxol-stabilised microtubules

*GMPCPP-microtubule seeds* (50% biotinylated, 50% ATTO-565 labelled tubulin) were prepared at a final concentration of 10 µM tubulin by incubating 5 µM biotin-labelled tubulin and 5 µM ATTO-565 labelled tubulin in BRB80 buffer supplemented with 1 mM GMPCPP for 1 h at 37 °C[51]. Microtubules were centrifuged at $85,000 \times g$ for 5 min at 37 °C, resuspended in the same volume of BRB80 buffer supplemented with 1 mM GMPCPP, aliquoted, and stored in liquid nitrogen.

*Taxol-stabilised microtubules* were polymerised by incubating 70 µM tubulin (50% biotinylated tubulin and 50% ATTO-565 labelled tubulin) in BRB80 buffer with 1 mM GTP for 45 min at 37 °C. 50 µM Taxol was then added and microtubules were incubated for 30 min before being centrifuged at $85,000 \times g$ for 5 min at 37 °C and resuspended in BRB80 buffer supplemented with 10 µM Taxol.

## Co-sedimentation assay

To assess the microtubule-binding properties of MAP6d1 and its mutants, 1 µM Taxol-stabilised microtubules were incubated for 25 min at 35 °C with 300 nM MAP6d1, MAP6d1-Δ2-35, MAP6d1-ΔMn2, MAP6d1-Mn2-7A or MAP6d1-Mn1-7A in BRB80 buffer supplemented with 125 mM KCl. The mixtures were then centrifuged at $190,000 \times g$ (Beckman Coulter rotor TLA-100) for 15 min at 35 °C. Protein contents of the supernatants and pellets were analysed by Western blotting using mouse monoclonal anti-tubulin (1:2000, clone alpha3A1)[53] and Penta His HRP-conjugated (1:10000, Qiagen, 34460) antibodies (Supplementary Figs. 3 and 4). The intensity of the bands corresponding to MAP6d1 and mutants was quantified using ImageJ.

## Microtubule dynamics using TIRF microscopy

Chambers made of glass slides and coverslips, functionalised with polyethylene glycol (PEG)-silane (30 Kda, Creative PEGWorks) and PEG-silane-biotin (3.4 Kda, Laysan BIO), respectively, were prepared as previously described[51]. Briefly, glass slides and coverslips were plasma-cleaned and immediately incubated overnight in the dark under gentle agitation with solutions of mPEG-silane (1 mg/mL in 96% ethanol, 0.2% HCl) and biotin-PEG-silane (1 mg/mL in 96% ethanol, 0.2% HCl), respectively. After incubation, slides and coverslips were thoroughly rinsed with ethanol and deionized water, dried, and assembled into flow chambers. The glass chamber was perfused with

neutravidin (25 µg/ml) (Pierce) in 1% BSA in BRB80, followed by Poly(L-lysine) (PLL)-g-PEG (0.1 mg/ml) (JenKem) in 10 mM Hepes, pH 7.4 and flushed with 1% BSA in BRB80. GMPCPP-microtubule seeds were then perfused and incubated for 5 min before being washed twice with 1% BSA in BRB80 buffer. Microtubules were polymerised from seeds by adding 12 µM tubulin (15% ATTO-565 labelled tubulin) with different concentrations of MAP6d1 or its mutants in TIRF buffer (BRB80 containing 1 mM GTP, 25 mM KCl, 3 mM dithiothreitol [DTT], 1% BSA, glucose [1 mg/ml], catalase [70 µg/ml], glucose oxidase [600 µg/ml] and 0.1% methylcellulose [4000 cP]). For experiments comparing MAP6d1 and mutants (Fig. 3a, b), the concentrations of MAP6d1-ΔMn2 and MAP6d1-Mn2-7A were increased to 300 nM because of their lower microtubule-binding activity relative to MAP6d1 (Supplementary Fig. 3b, c). Flow chambers were sealed, and movies were acquired on an inverted microscope (Eclipse TI, Nikon) equipped with an iLas TIRF system (Roper Scientific), a CMOS camera (Prime 95B, Photometrics) and a temperature-controlled stage (LINKAM MC60), under MetaMorph software (version 7.7, Molecular Devices). Samples were excited using a 561-nm laser and observed with an Apochromat 100x oil-immersion objective (N.A., 1.49). Time-lapse acquisitions were carried out for 30 min at 35 °C with a 100 ms exposure and one frame every 2 s.

To assess the effect of dilution of microtubule stability, microtubules were polymerised in glass chambers from ATTO-565-GMPCPP seeds in the presence of 12 µM tubulin (15% ATTO-565 labelled tubulin) and 50 nM MAP6d1 in TIRF buffer containing 1 mM GTP. For control conditions, microtubules were polymerised from 25 µM tubulin in TIRF buffer with 1 mM GTP. Unsealed chambers were then incubated for 20 min at 35 °C in a humid environment. Chambers were then placed on the warm stage of the TIRF microscope and microtubule depolymerisation was induced by flowing warm TIRF buffer. Microtubule behaviour was immediately recorded using 561-nm laser and images were acquired every 2 s with a 100 ms exposure for 30 min at 35 °C.

To evaluate the dynamic behaviour of microtubules polymerised at the extremity of MAP6d1-assembled-microtubules, microtubules were grown from ATTO-565-GMPCPP seeds in the presence of 12 µM tubulin (15 % ATTO-565 labelled tubulin) and 50 nM MAP6d1 in TIRF buffer supplemented with 1 mM GTP. The unsealed chambers were incubated for 20 min at 35 °C in a humid environment, before perfusion of a solution containing 12 µM tubulin with 1 mM GTP in TIRF buffer. Images were then acquired under the microscope every 2 s with 100 ms exposure for 30 min at 35 °C.

## Tubulin recruitment assay

Mixtures containing 0.325 µM tubulin (15 % ATTO-491 labelled tubulin) and either MAP6d1 (100 nM), MAP6d1-Δ2-35 (100 nM), MAP6d1-ΔMn2 (300 nM), MAP6d1-Mn2-7A (300 nM) or MAP6d1-Mn1-7A (300 nM) in TIRF buffer with 50 µM GMPCPP and 50 mM KCl were incubated with ATTO-565-GMPCPP seeds in chambers. Samples were excited using 491- and 561-nm lasers, and time-lapse images were acquired on the TIRF microscope every 2 s with 100 ms exposure for 30 min at 35 °C.

## TIRF analysis

Movies were analysed using ImageJ software to generate kymographs of individual microtubules. The four parameters to assess microtubule dynamics were extracted from kymographs using an in-house plugin[51]. Growth and shrinkage rates were calculated from the slopes of the corresponding growth and shrinkage events, respectively. Catastrophe and rescue frequencies were calculated for each microtubule by dividing the number of events by their corresponding growth or shrinkage event durations. Microtubule pauses were defined by growth/shrinkage rates less than 0.24 µm/min at the plus end and 0.12 µm/min at the minus end.

## Negative staining electron microscopy

We used two protocols to evaluate microtubule doublet formation. In one, we co-polymerised 25 µM purified brain tubulin with 250 nM MAP6d1 in BRB80 buffer supplemented with 1 mM GTP and 125 mM KCl for 10 min at 35 °C (Fig. 2a). In the second approach, we incubated 10 µM stabilised GMPCPP-microtubule seeds with purified brain tubulin (0.325 µM) and either MAP6d1 (100 nM), MAP6d1-Δ2-35 (100 nM), MAP6d1-ΔMn2 300 nM), MAP6d1-Mn2-7A (300 nM) or MAP6d1-Mn1-7A (300 nM) in BRB80 buffer supplemented with 50 µM GMPCPP and 50 mM KCl for 10 min at 35 °C (Fig. 2e and Supplementary Fig. 4e). We then loaded 4 µL of each sample on a 400-mesh copper carbon grid (Electron Microscopy Sciences), washed it briefly with warm BRB80 buffer and stained it with 2.5% uranyl acetate. Images were acquired using a transmission electron microscope (TEM) JEOL 1200 EX (Veleta). The length of microtubule doublets and singlets was measured on images using ImageJ software.

## Cryo-electron tomography

**In vitro sample preparation.** 25 µM tubulin was co-polymerised with 250 nM MAP6d1 supplemented with 1 mM GTP and 125 mM KCl in BRB80 buffer for 5–10 min at 35 °C. Four microliters of the solution was mixed to 1 µL of cationic BSA-coated gold beads (Aurion Gold Tracers, 210111) and then loaded on a glow-discharged Quantifoil R2/2 300 mesh copper grid (Electron Microscopy Sciences) and placed in the Leica EM-GP2 95 humidity chamber. The grid was blotted for 2 s and plunge-frozen in liquid ethane.

**Neuronal sample preparation.** Glow-discharged Quantifoil R2/2 200 mesh gold grids (Electron Microscopy Sciences) were placed in poly-L-lysine-coated sterile 35-mm glass-bottom dishes (Ibidi, FluoroDish) and hippocampal neurons grown on the above-mentioned grids for 25 days, following the protocol described for neuronal culture. The grid was blotted for 5 s using the vitrification robot Leica EM-GP2 and plunge-frozen in liquid ethane.

**Image acquisition.** The tilt series were acquired using K3 Titan Krios at the ESRF facility[54] operated at 300 keV, at a magnification of ×33,000 between −60° and +60° with a 3° increment under an electron dose of 2.87 e/Å$^2$ at a defocus of −3 to −6 µm, using Tomography 5 software (version 5.16, Thermo Fisher Scientific). The acquisition was done with a super-resolution pixel size of 1.35 Å/px and the data processing was carried out in binning 2 with a pixel size of 2.7 Å/px (see Supplementary Table 2).

**Image processing.** The 252 and 118 tilt series composed of 41 images were acquired for the in vitro samples and neurons, respectively. Raw tilt images were motion-corrected using MotionCor2 (version 1.4.0)[55] and then aligned for reconstructing the 3D tomograms by automatic alignment followed by contrast transfer function (CTF) estimation using the tomography pipeline in Eman2 software (version 2.99)[56] (Supplementary Fig. 7 and Supplementary Table 2).

For in vitro samples, we manually selected different microtubule architectures (singlet and doublet microtubules, with or without luminal protofilaments, see Supplementary Fig. 2a, b) using the filament tracing tool and particles were extracted with periodicities of 8 nm. The number of protofilaments in singlet microtubules and the A-tubule of doublet microtubules was determined on extracted particles by performing 3D classification in Eman2, using microtubules with 11–16 protofilaments as references. For the control, protofilament analysis was performed based on the microtubule fringe pattern on 2D cryo-EM images[57]. In the presence of MAP6d1, singlet microtubules and the A-tubule of doublet MTs consistently contained 14 protofilaments (Supplementary Fig. 2c).

For doublet microtubule reconstruction, 4997 particles were selected. First, we built reference models of doublet microtubules consisting of an A-tubule with 14 protofilaments and a B-tubule with variable numbers of protofilaments (Fig. 2b). We then classified the particles based on the number of protofilaments in the B-tubule, using Eman2's multi-reference refinement. The distribution of particles with 7, 8, 9, 10, or 11 protofilaments in the B-tubule made up 21.27, 11.28, 9.85, 39.92 and 17.67%, respectively (Supplementary Fig. 2c). Sub-tomogram averaging refinement was performed on all extracted particles. Since the largest class (39.92%) corresponded to doublet microtubules with a 14-protofilament A-tubule and a 10-protofilament B-tubule, we used this configuration as the reference for the 3D-refinement step applying Eman2's 3D refinement pipeline that includes iterative steps of 3D particle alignment, sub-tilt translational and rotational refinements followed by defocus variation and using a mask including both A- and B-tubule. We then applied a local mask to the B-tubule to refine it independently of the A-tubule. The map displayed is the combination of both maps (Fig. 2d and Supplementary Fig. 2d).

We selected 5881 and 1680 particles, respectively, to reconstruct singlet and doublet microtubules with luminal protofilaments. We performed subtomogram averaging using Eman2's 3D refinement pipeline as described above. We verified that all singlet microtubules contained two luminal protofilaments through 3D classification using 14-protofilament microtubules with either one or two luminal protofilaments as references. To improve the reconstruction of the luminal protofilaments in singlet microtubules, we applied a local mask to the two luminal protofilaments to refine them independently of the microtubule lattice (Fig. 5c).

To visualise luminal protofilaments on tomograms of mature neurons, we searched for structures similar to those identified in vitro, i.e., filamentous densities close to the microtubule wall and/or displaying curved extremities. Note that the observation of these events is limited by the low signal-to-noise ratio of tomograms due to the thickness of the sample and the presence of numerous MIPs in the microtubule lumen, which may obscure elongated luminal structures.

The IMOD package (version 4.11.25) was used for displaying tomograms[58]. Electron density maps were visualised with Chimera (version 1.19) and ChimeraX (version 1.2.5)[59]. FSC plots were extracted from Eman2 refinement output (Supplementary Fig. 2e).

## B-tubule curvature analysis

To measure the curvature radii of B-tubules (related to Figs. 2 and 5), we selected a set of four doublet microtubules without luminal protofilaments and three doublet microtubules with luminal protofilaments. To ensure that curvature measurements were not influenced by neighbouring structures, we chose only doublet microtubules far from the carbon support film and well isolated from each other. Curvature was measured at nine different z-sections along each doublet. For each z-section, about ten coordinates were selected to trace the B-tubule curve using the segmented line tool in Fiji software[60] (version 2.160/1.54p). To superimpose the resulting B-tubule traces, we defined two reference points: the centre of the A-tubule (O) and the outer junction where the B-tubule attaches (X). Each set of the z-section coordinates was then rotated and translated such that all O-X segments were aligned and all X points were centred at the same position. Mean curvature radii were calculated along each B-tubule trace using the curvature_radius.bsh script, originally written by Olivier Burri (BIOP EPFL) and modified for our study (https://github.com/EricDenarier/MAPd1_MT_Doublets_2025), and the results were plotted on graphs.

The mean curvature radius of the B-tubule in in vivo doublet microtubules was determined using Fiji by drawing two circles on a z-stack of the *Chlamydomonas* flagellar doublet microtubule (EMD-40621)[17], one corresponding to the external surface of the tubulin, the

other to its internal surface. The final radius corresponds to the average value calculated from the radii of these two circles.

## Animals used in this study

The *Map6d1* conditional KO mutant mouse line was established at the Institut Clinique de la Souris (ICS) - PHENOMIN. The targeting vector was constructed as follows: A 3 kb 3′ homology arm fragment was amplified by PCR (from 129svpass genomic DNA) and subcloned in an ICS proprietary vector. This ICS vector bore a floxed and flipped Neomycin resistance cassette and a 5′ LoxP site. A 1.9 kb fragment encompassing 628 bps of proximal promoter and *Map6d1* exon 1 (ENSMUSE00000268483) was cloned in a second step. Finally, a 4.2 kb fragment corresponding to the 5′ homology arms was amplified by PCR and subcloned in step 2 plasmid to generate the final targeting construct. The linearised construct was electroporated in 129svpass (ICS derived P1 line) mouse embryonic stem (ES) cells. After G418 selection, targeted clones were identified by 5′ and 3′ Long 6range PCRs and further confirmed by Southern blot with a Neo probe (5′ and 3′ digests). Three positive ES clones were injected into C57BL/6 J blastocysts. Chimeras were bred with Flp deleter females (in a C57BL/6 J genetic background) in order to obtain the cKO allele with no selection cassette. Germline transmission was achieved.

To obtain the plain KO allele, chimeras were bred with Cre deleter females (also in a C57BL/6 J genetic background) (Supplementary Fig. 8). Genotyping was performed by PCR on genomic DNA isolated from mouse tails using following oligonucleotides: 5′-GG GGCTTATGCCTGTGGCTATATG-3′ plus 5′-CAGTCCTCCTAGGTGCAGA CTG-3′ to detect the mutant allele and 5′-CATGGATTCCAGGGATCCA TCTCTC-3′ to detect the wild-type allele. All experiment were performed on a C57BL6/J genetic background. Mice were house in a temperature controlled ($22 \pm 1\,°C$) under 12:12 light/dark cycle (light from 7 A.M. to 7 P.M.) with *ad libitum* access to food and water and humidity between 45 and 65%.

## Cultured hippocampal neurons

Hippocampal neurons were isolated from the brains of wild-type and *MAP6d1*-knockout (*MAP6d1*-KO) littermate embryos (of both sex) at embryonic day 17.5 (E17.5) and cultured as described by Gory-Fauré et al.[61]. Briefly, the brains were dissected, and the hippocampi were removed, dissociated, and plated on poly-L-lysine-coated coverslips in 35 mm dishes containing DMEM 10% horse serum. After 2 h 30 min, the culture medium was replaced with MACS medium.

In some experiments, neurons were transfected prior to plating using the Lonza Mouse Neuron Nucleofector 2b Kit (program 0-005, following the manufacturer's instructions). Neurons were electroporated with 4 µg of plasmid encoding control GFP, MAP6d1-GFP, or MAP6d1-GFP deletion mutants immediately following dissociation. To ensure efficient transfection, neurons from one hippocampus were plated per dish.

## Immunofluorescence microscopy and quantification

Cells were fixed in 4% paraformaldehyde, 4% sucrose in phosphate buffered saline (PBS) for 30 min at 37 °C, followed by permeabilisation in 0.2% Triton X-100 in PBS for 2 min at room temperature. Cells were then incubated with primary antibodies for 1 h at room temperature. Coverslips were washed 3 times in PBS - 0.1% Tween 20 and incubated with conjugated secondary antibodies for 1 h in the dark at room temperature. After three final PBS 0.1% Tween20 washes and one with PBS, nuclei were stained using Hoechst (Sigma) in the mounting medium (Dako).

Primary antibodies used included mouse monoclonal anti-Arl13b (1:1000, Addgene, 180085), mouse monoclonal anti-polyglutamylated tubulin (1:1000, Coger, AG-20B-0020), rabbit polyclonal anti-acetylated tubulin (1:5000, Millipore, ABT241), and rat monoclonal anti-tyrosinated tubulin (1:500, clone YL1/2[62]). The secondary antibodies were donkey anti-mouse Cy3 (715-165-151), donkey anti-rabbit AF647 (711-605-152) and donkey anti-rat AF488 (712-545-153) (1:500, Jackson Immuno-Research Laboratory).

Images of cilia were acquired on a Zeiss LSM 710 confocal microscope (equipped with a Zeiss AiryScan module), using a 63x oil-immersion NA 1.4 objective and Zen software (Zen 2.1 SP3 FP3 version 14.0.23.201, Carl Zeiss MicroImaging).

We measured ciliary length using Fiji on the maximal projection of confocal images of cilia, all combined in a single file to facilitate manipulation. Only cells with a visible GFP signal at the Golgi and neurites were included (Supplementary Fig. 5c). We traced the cilia manually to determine their lengths using the segmented line tool.

For fluorescence intensity measurements, we imported confocal images into Fiji and max-projected the z-planes spanning the cilium. The entire cilium was manually traced from the most proximal end to the distal end with a width of 0.17 µm. We then extracted the intensity profile of the selected region, normalising length and intensity to a 0-100 scale. To compare MAP6d1-GFP localisation to polyglutamylated tubulin, we drew a line along acetylated tubulin ciliary staining and compared the signal of both proteins along this line. Similarly, we drew a line along Arl13b ciliary staining to compare the respective protein signals of MAP6d1-GFP and tyrosinated tubulin localisation.

## Expansion microscopy

At DIV9, five days after transduction with a lentivirus expressing MAP6d1-EGFP, neurons were fixed in 4% paraformaldehyde and 4% sucrose in phosphate-buffered saline (PBS) for 30 min at 37 °C. Following fixation, neurons were stored in PBS for 4 days prior to processing for post-expansion staining, following the protocol from Gambarotto et al.[63]. Briefly, coverslips were incubated for 5 h at 37 °C in a solution of 2% acrylamide and 1.4% formaldehyde in PBS. Polymerisation was then initiated by incubating the samples for 1 h at 37 °C in a monomer solution containing 19% (w/v) sodium acrylate, 10% acrylamide, 0.1% bis-acrylamide, 0.5% (v/v) TEMED, and 0.5% (w/v) ammonium persulfate in PBS. This was followed by a denaturation step at 95 °C for 1 h and 15 min.

Following the first round of expansion, gels were immunostained with the following primary antibodies: rabbit anti-GFP (1:500, Life Technologies, A11122) and mouse anti-adenylate cyclase (1:250, Life Technologies, MA5-47381) or mouse anti-polyglutamylated tubulin (1:250, Coger clone GT335). The secondary antibodies were donkey anti-mouse Cy3 and donkey anti-rabbit AF488 (711-545-152) (1:500, Jackson Immuno-Research Laboratory). Images were acquired using a Zeiss LSM900 confocal microscope equipped with a 63x, NA 1.3 objective in Airyscan mode. To isolate the ciliary structure and reduce adjacent labelling, cilia outlined from different z-planes were extracted using the BigTrace plugin for Fiji (https://github.com/ekatrukha/BigTrace). A maximal intensity projection is presented in Fig. 4d, e.

## Statistical analysis

All experiments were repeated at least three times. Statistical analysis was performed with Prism GraphPad version 10. Statistical tests and sample sizes are all indicated in the figure legends.

All in vitro experiments conducted with MAP6d1 and its mutants are summarised in Supplementary Table 3, including the concentrations used and the results obtained.

## Reporting summary

Further information on research design is available in the Nature Portfolio Reporting Summary linked to this article.

# Data availability

All data supporting the results of this study can be found in the article, Supplementary Information, and Source Data files. Materials can be

obtained from the corresponding authors under a material transfer agreement. The subtomogram average maps data in this study have been deposited in the EMDB under the following accession codes: EMD-52569 (singlet microtubule with luminal protofilaments), EMD-52572 (doublet microtubule with luminal protofilaments), EMD-52575 (doublet microtubule), EMD-53457 (combined map of singlet microtubule with luminal protofilaments, from focused refined maps EMD-53454 (singlet microtubule) and EMD-53452 (luminal protofilaments)), EMD-52623 (combined map of doublet microtubule from focused refined map EMD-52574 (B-tubule)). Source data are provided with this paper.

## Code availability

We have used the curvature_radius.bsh script, originally developed by Olivier Burri (BIOP, EPFL) and modified for our study to analyze the curvature of B-tubule traces and calculate their mean curvature radii, which is available via GitHub [https://github.com/EricDenarier/MAPd1_MT_Doublets_2025].

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

## Acknowledgements

We thank N. Chaumontel and C. Miscopein for helping protein purification and cell biology experiments, M. Benoît for helpful discussion and V.L. Brandt for editing the manuscript. This work used the Photonic Imaging Center of Grenoble Institute of Neuroscience (PIC-GIN, University Grenoble Alpes – INSERM U1216), which is part of the ISdV core facility and certified by the IBiSA label. We also used the two EM facilities in Grenoble: the European Synchrotron Radiation Facility (beam time on CM01) and the Grenoble Instruct-ERIC Center (ISBG; UMS 3518 CNRS CEA-UGA-EMBL) with support from the French Infrastructure for Integrated Structural Biology (FRISBI; ANR-10-INSB-05-02) and GRAL, a project of the University Grenoble Alpes graduate school (Ecoles Universitaires de Recherche) CBH-EUR-GS (ANR-17-EURE-0003) within the Grenoble Partnership for Structural Biology. The IBS Electron Microscope facility is supported by the Auvergne Rhône-Alpes Region, the Fonds Feder, the Fondation pour la Recherche Médicale and GIS-IBiSA. We thank the staff members of PIC-GIN and both EM facilities for their assistance. The mice used in this study were bred at the animal facility of IRIG-DRF-CEA (Grenoble, France), which is supported by funding from GRAL, a program of the University Grenoble Alpes graduate school (Ecoles Universitaires de Recherche) CBH-EUR-GS (ANR-17-EURE-0003). We thank the zootechnicians, S. Bama-Toupet and C. Magallon, for their assistance. This work was supported by the Institut National pour la Santé et la Recherche Médicale (INSERM), the Centre National de la Recherche Scientifique (CNRS) and l'Agence Nationale pour la Recherche (ANR-20-CE13-0005-01 to A.A., ANR-22-CE16-0018-01 to A.A., ANR-24-CE13-5131-01 to I.A.). D.G. was supported by ANR (20-CE13-0005-01), J.W. by the French MESR Ministry and M. de A. by ANR (ANR-22-CE16-0018-01).

## Author contributions

Conceptualisation: S.G.-F., L.S., I.A. Investigation: D.G., J.W., J.D., C.B., M. de A., G.E., E.D., S.G.-F., L.S. Formal analysis: D.G., J.W., J.D., M. de A., E.D., S.G.-F., L.S., I.A. Resources. G.E., E.D. Writing - Original draft: D.G, L.S., I.A. Writing - review and editing: A.A., D.G, J.W., E.D., G.E., C.B., S.G-F, L.S., I.A. Visualisation: D.G., J.W. Supervision: S.G-F, L.S., I.A. Funding acquisition: A.A., I.A.

## Competing interests

The authors declare no competing interests.
