## [Transparent Peer Review file · Nature Communications]

The Mn-motif protein MAP6d1 assembles ciliary doublet microtubules

Corresponding Author: Dr Isabelle Arnal

Version 0:

Reviewer comments:

Reviewer #1

(Remarks to the Author)

The publication entitled “The Mn-motif protein MAP6d1 assembles ciliary doublet microtubules” shows convincingly that MAP6d1 can assemble ciliary doublets in vitro and that its activity is dependent on the second Mn motif as well as its N-terminal region. The authors also show that MAP6d1 is essential for proper ciliogenesis in neurons. The manuscript will be of interest to the microtubule field, specifically people studying microtubules in neurons as well as cilia.

On the other hand, the manuscript still needs revision in order to make it appropriate for publication.

Comments:

The concentrations used of tubulin and protein seem to be different depending on the experiment. Fig. 1 uses 12 μM tubulin and up to 50 nM protein and no tubulin recruitment is observed in Fig. 1g. Fig. 2a shows a quantification of doublet formation with 25 μM tubulin and 250 nM MAP6d1. Fig. 2e shows recruitment of tubulin when 0.325 μM tubulin and 100 nM protein are used. Finally, Supplementary figure 1 shows a co-sedimentation assay with 1 μM tubulin and 300 nM protein.

I believe the reader would benefit from seeing these numbers directly on the figure. Alternatively, a summary table with the different conclusions obtained with the different concentrations of tubulin and protein could be shown, stating if there is doublet formation and/or tubulin recruitment and/or microtubule stabilisation at each given concentration.

In line with the previous comments, the methods section says that protein concentration “was measured using BSA as standard by loading onto an SDS-PAGE gel”. It is not very clear to me how the authors measured the protein concentrations.

I would argue that protein concentration needs to be determined accurately as the experiments use very small amounts of protein. Furthermore, the proteins that are purified appear to have a contaminant at $\sim 45\text{kDa}$. Have the authors tried size exclusion chromatography in order to obtain a “cleaner” protein?

Furthermore, all proteins have a 8-His tag which is known to, in some cases, help proteins bind to microtubules (see

<https://www.sciencedirect.com/science/article/pii/S0021925820378571?via=ihub> and

<https://www.science.org/doi/10.1126/sciadv.abq3817>). Have the authors tried similar experiments with proteins that did not have a His tag?

The observation of the authors regarding the doublet and singlet with luminal pf looks convincingly consistent. But unfortunately, it is still not clear whether the seam is located at the designated PF A9 & A10 as indicated by the authors. Does that mean Map6d1 has seam recognition property? Any inside from Mn1 domains in the MIP regarding this? In addition, using kinesin decoration on the cryo-ET data might help to clear that out in the case of singlet-luminal PF. For the doublet, it might be more difficult if good decoration is not achieved.

The observation of single protofilaments inside singlet microtubules in vivo is new and exciting but is not very convincing. Currently, the figure looks to me as though it could be a long filamentous protein as the tektin bundles inside the sperm doublet microtubule. Could the authors provide a gallery of these observations as well as a quantification of the occurrence of protofilaments inside microtubules? Could the authors try subtomogram averaging to strengthen their claim? Although the authors do not claim that MAP6d1 is needed for luminal protofilaments in vivo, it would be interesting to know if similar observations can be made in MAP6d1-KO cells.

In addition, the quality of visualization for in vivo tomograms would improve a lot if a denoising program is used such as IsoNet or CryoCare or DeepDeWedge.

In addition, image processing must be better described. There are a lot of details missing in the image processing. Perhaps a diagram showing the processing workflow is better describe it. Is the search used filament properties to limit the search angles? It was puzzling to me that the maps provided by the authors have a very weird orientation. Normally, when you have a filament, the filament axis must be along the Z axis. That allows better comparison of the authors' structure with other structures.

There are not clear how the singlets with different protofilament numbers are classified, how do the singlets with luminal pf are aligned, and any variability in the number of luminal pfs (from the authors' image, I could see that might be 1 or 2 pfs). Does the author do focus refinement on the luminal space, allowing better resolution of the luminal protofilament and more confident fitting?

Supplementary Fig. 1 shows that a mutant lacking the second Mn2 domain has a greatly reduced affinity for microtubules. What is the role of the first Mn2 domain? Would it make sense to create a mutant of the first Mn2 domain and study its effects on microtubule stability?

I think Fig. 3 needs a more complete description along with more statistics in panel b to compare the mutants. If the conclusion is that the deltaMn mutant and the Mn-7A don't stabilise microtubules, then one would expect no significant difference between mutants and control MTs. MAP6d1 delatN stabilises microtubules and reduces catastrophe frequency while the WT seems to increase it. The same mutant also seems to increase the growth rate. Could these observations suggest an additional role for the N terminus to its requirement for microtubule pausing?

Minor suggestions

Page 3: Cite Supplementary Fig. 1 before Fig.2

Page 4: Cite Supplementary Fig. 2b before Supplementary Fig. 3.

Page 6: Ref 41 is for mitosis.

Page 6: ...adjacent to the putative seam (Fig 5c) -> Fig 5b should be cited

Page 8: KIF21B: ref 45 should be cited

Fig 1: panel a, seed should be moved to the left

Fig 2c: very pixelated

Fig 4: "stained for" instead of "stained against"

Fig 5: write "putative seam" instead of seam in Figure 5

Fig S1: Remove extra WB, add MW next to ladder.

Fig S2a: catastrophe frequency, the bar for the mean is missing for 20nm

Fig S4a legend: circles are the same colour – also true for other figure legends.

Reviewer #2

(Remarks to the Author)

Reviewer #3

(Remarks to the Author)

In this manuscript, Gopal and colleagues explore the functions of MAP6d1 protein, also called SL21 (STOP-like 21), member of the Stable Tubule Only Polypeptide (STOP) family, and the Stabiliser of AXOnemal microtubules (SAXO) family.

Using TIRF microscopy and cryo-EM of the in vitro reconstituted components, the authors show that MAP6d1 is a potent MT stabilizer which induces pauses of MT growth/shrinkage, and that it induces MT doublet formation based on its ability to concentrate soluble tubulin on the lattice of single MTs. They further show that these properties are dependent on the Mn2 and N-terminal domains of MAP6d1. CryoEM of microtubules in vitro allows authors to demonstrate the presence of protofilaments present inside single or doublet MTs, a phenomenon that was never observed before and that might change our view on microtubule biology. One possible role of this filament might be to determine the localisation of the B-tubule, an exciting hypothesis the authors discuss. Moreover, the authors also observe intra-luminal protofilaments in MTs in primary neurons, providing evidence for the presence of this phenomenon in vivo. The authors suggest that these filaments could confer stability to neuronal, long-lived MTs. Finally, they demonstrate that in primary mouse neurons, MAP6d1 - localised to the proximal part of the cilium - is necessary for cilia length control.

The manuscript is clear and well-written, the experiments are thoroughly described, and figures are assembled with care and are easy to understand. Analyses are meticulous and well documented.

The scientific question is interesting and timely, as the research into MIPs is still in its early stages. Furthermore, the described effects of MAP6d1 on the formation of intra-luminal protofilaments both in vitro and in cultured neurons are fascinating and new and could add to our understanding of how neuronal microtubule stability is achieved. I believe that a better effort could be made to better distinguish between what MAP6d1 can do in vitro, and what really is its function in vivo, which is why I have a few questions/remarks, that would be important to address before the manuscript can be accepted for

publication.

1. I believe it is important to distinguish what a protein can do in an in vitro setting, and what it really does in cells. In this respect, how do the authors reconcile the conclusion that MAP6d1 is important for cilia development, as cilia seem to develop before MAP6d1 is even expressed in mice? Accordingly, in DIV4 neurons there is no difference between cilia in WT and MAP6d1KO neurons. Could it rather be that MAP6d1 in vivo stabilizes cilia, rather than contributes to their development/building? Please revise the discussion accordingly.

2. In the buffer perfusion experiments (Fig1d): is MAP6d1 present in the perfused buffer? If not, does it not get diluted out?

3. Where do the percentages given while discussing Fig 3c/d come from (1%-40% and 9% - 57%)? I can't see those values in the graphs.

4. Fig 4: can the authors show by EM that MAP6d1 really colocalizes with the doublet MTs in the cilium? Do you predict that in DIV9 MAP6d1-KO cilia there are no doublet MTs left, while after MAP6d1- Δ 2-35 overexpression the doublet MTs go up to the tip of the cilium (Fig 4c)?

5. Fig 5: Did the authors ever observe intraluminal protofilaments when MAP6d1 is added to already-polymerised MTs?

6. Fig 5: How were the radii of the B-tubules measured? What is the radius of the B-tubule in vivo? It would be interesting to compare this value to the values measured in vitro.

Textual changes:

1. Please add in the introduction the alternative names of MAP6d1.

2. The sentence p.2: MAP6, a microtubule-stabilising factor linked to psychiatric disorders^{24,25}, was the first neuronal MIP located in the microtubule lumen, where it generates highly stable microtubules that grow in a helicoidal pattern²⁶. implies that MAP6 induces the growth of helicoidal MTs inside the lumen of the MT lumen: please rephrase.

3. In the same paragraph the word "motif" is repeated 4 times.

4. Title of the first results chapter: MAP6d1 stabilises microtubules by inducing pauses: pauses of what?

5. Last paragraph p.5: for ectopic expression, please add information about time of transfection and time of analysis. Shouldn't " Δ Mn2-35" be " Δ 2-35"?

6. 2nd paragraph p.6: "performed immunofluorescence co-staining" not "co-immunofluorescence staining".

7. Last paragraph of p.6: remove "of doublet MTs".

8. 2nd paragraph of the discussion, 2nd sentence: a definition of a MAP does not include to be able to "recruit tubulin on the MT lattice", just like the definition of a MIP is not to "form intraluminal protofilaments". Please rephrase.

Version 1:

Reviewer comments:

Reviewer #1

(Remarks to the Author)

In my opinion, the author did a thorough job of addressing all the comments and suggestions from the reviewers. Especially, using the polyE antibody to mark the doublet microtubule, showing that MAP6d1 localizes to the doublet microtubule region of the primary cilia. Therefore, the paper should be ready to publish.

Upon reading the manuscript again, I have some minor points that the author can address quickly before publication.

In their in vitro dataset, does the author encounter the doublet microtubule with more than 1 B-tubule (or more than 1 B-tubule hook)? From their dataset, DMT is about 20%. Therefore, the chance of having two hooks should be 4% if the hooks form independently. So, if the authors see no more than 1 hook in their dataset, there is a likelihood that MAP6d1 can be very specific to the 'putative seam'. So, a calculation and a statement on this in the paper would be good.

Is the subtomogram refinement Gold Standard Refinement (divided into two independent half sets and refined)? The reason is that if you use Gold Standard Refinement, then the FSC should be calculated at 0.143 not 0.5. FSC=0.5 is used in Non-Gold Standard Refinement. If Non-Gold Standard Refinement is used, the FSC should be clearly stated as Non-Gold Standard FSC or Semi-Independent FSC.

Reviewer #2

(Remarks to the Author)

Reviewer #3

(Remarks to the Author)

I thank the authors for having addressed all my comments. The paper can be published in the current state.

We thank the reviewers for their thoughtful appreciation of our work and detailed suggestions for clarifying and strengthening the manuscript. We have done our best to accommodate their comments. Below, we provide a point-by-point response. For clarity, we quote the original reviewer comments in *italics*, and our responses are in regular font.

Reviewer #1 (Remarks to the Author)

The publication entitled “The Mn-motif protein MAP6d1 assembles ciliary doublet microtubules” shows convincingly that MAP6d1 can assemble ciliary doublets in vitro and that its activity is dependent on the second Mn motif as well as its N-terminal region. The authors also show that MAP6d1 is essential for proper ciliogenesis in neurons. The manuscript will be of interest to the microtubule field, specifically people studying microtubules in neurons as well as cilia.

On the other hand, the manuscript still needs revision in order to make it appropriate for publication.

We thank the reviewer for their careful consideration and constructive suggestions.

Comments:

The concentrations used of tubulin and protein seem to be different depending on the experiment. Fig. 1 uses 12 μ M tubulin and up to 50 nM protein and no tubulin recruitment is observed in Fig. 1g. Fig. 2a shows a quantification of doublet formation with 25 μ M tubulin and 250 nM MAP6d1. Fig. 2e shows recruitment of tubulin when 0.325 μ M tubulin and 100 nM protein are used. Finally, Supplementary figure 1 shows a co-sedimentation assay with 1 μ M tubulin and 300 nM protein.

I believe the reader would benefit from seeing these numbers directly on the figure. Alternatively, a summary table with the different conclusions obtained with the different concentrations of tubulin and protein could be shown, stating if there is doublet formation and/or tubulin recruitment and/or microtubule stabilisation at each given concentration.

This was an excellent suggestion. We have added a new Supplementary Table 1 summarizing the conclusions from the various experiments, along with the corresponding tubulin and protein concentrations used in each case.

In line with the previous comments, the methods section says that protein concentration “was measured using BSA as standard by loading onto an SDS-PAGE gel”. It is not very clear to me how the authors measured the protein concentrations. I would argue that protein concentration needs to be determined accurately as the experiments use very small amounts of protein.

We agree, and apologize for not being more clear. In our study, we used a commercial BSA standard with a precisely known initial concentration. This was most appropriate in our case, as our sample is not 100 % pure, as highlighted by the reviewer, and BSA is commonly used as a standard in concentration measurement methods such as Bradford or Lowry. The intensity of the band corresponding to our protein of interest was compared to the intensities of different BSA standard dilutions loaded on the same gel. Band intensities were quantified using ImageJ. We have clarified this point in the Material and Methods section of the revised version.

Furthermore, the proteins that are purified appear to have a contaminant at ~45kDa. Have the authors tried size exclusion chromatography in order to obtain a “cleaner” protein?

We tried adding a gel filtration step to improve protein purity, but significant protein loss at this stage led us to exclude this additional step from the final protocol.

Furthermore, all proteins have a 8-His tag which is known to, in some cases, help proteins bind to microtubules (see <https://www.sciencedirect.com/science/article/pii/S0021925820378571?via=ihub>

and <https://www.science.org/doi/10.1126/sciadv.abq3817>). Have the authors tried similar experiments with proteins that did not have a His tag?

We did not attempt to purify proteins without a His-tag, which would have significantly complicated the purification process. Nevertheless, we have good reason to believe that the His-tag does not interfere with microtubule binding in our study:

Firstly, all proteins used in this study (wild-type and mutants) carry the same C-terminal 8xHis-tag, yet they exhibit different behaviors in terms of microtubule interaction. Specifically, the MAP6d1- Δ -Mn2 mutant shows weak interaction, while MAP6d1 and MAP6d1- Δ 2-35 display stronger binding. These differences indicate that the His-tag is not the determinant in the microtubule binding of our proteins.

Secondly, at a pH of 6.8 (the pH of all tested conditions), half of the histidines are deprotonated and thus neutral. This greatly reduces potential electrostatic interactions between the His-tag and the microtubule surface.

Thirdly, the use of KCl in all our experiments, particularly at 125 mM in the conditions used for EM, also prevents non-specific interactions between MAP6d1 proteins and microtubules.

The observation of the authors regarding the doublet and singlet with luminal pf looks convincingly consistent. But unfortunately, it is still not clear whether the seam is located at the designated PF A9 & A10 as indicated by the authors.

This is a fair point. We postulated that the seam is located between protofilaments A9 and A10 in the doublet microtubule based on previous structural studies (Ichikawa et al., *Nat Comm*, 2017; Ma et al., *Cell*, 2019). We have now updated Figure 5, replacing “seam” with “putative seam” and have added these references in the text when mentioning the seam (p. 7).

Does that mean Map6d1 has seam recognition property? Any inside from Mn1 domains in the MIP regarding this?

To answer these questions, it would be necessary to determine the high-resolution structure of MAP6d1 bound to singlet and doublet microtubules. To our knowledge, no structural data has demonstrated that Mn motifs alone confer specific binding to the seam. In fact, Andersen et al. (*Nat Comm*, 2023) showed that among the MIPs containing Mn motifs, four—CFAP68, CFAP95, CFAP107 and SPAG8—actually bind the seam via their NWE motif, which the MAP6d1 sequence lacks.

In addition, using kinesin decoration on the cryo-ET data might help to clear that out in the case of singlet-luminal PF. For the doublet, it might be more difficult if good decoration is not achieved.

The reviewer is correct about both the approach and its potential difficulty. Unfortunately, it would take considerable time to develop this approach, which puts it beyond the scope of the present study. We hope to come back to this at some point in the future.

The observation of single protofilaments inside singlet microtubules in vivo is new and exciting but is not very convincing. Currently, the figure looks to me as though it could be a long filamentous protein as the tektin bundles inside the sperm doublet microtubule.

In the original text (p. 7), we were careful to describe our observation with due circumspection: “We found singlet microtubule structures containing filamentous densities *that resemble* the luminal protofilaments induced *in vitro* by MAP6d1 in neuritic extensions.” Nevertheless, our *in vitro* data suggests that these luminal filaments could correspond to tubulin protofilaments, which is supported by our *in vivo* identification of similar configurations, specifically filamentous densities positioned close to the microtubule wall and/or displaying curved extremity—features reminiscent of those observed in microtubules polymerized with MAP6d1 *in vitro*.

We do not favor the hypothesis that the filaments inside singlet neuronal MTs are tektin filaments. To the best of our knowledge, tektin bundles have been observed only in the doublet microtubules of cilia and flagella, not in cytoplasmic singlet MTs or the singlet MTs found at the tip of cilia (Legal et al, *BioRxiv* 2024; Amos and Amos, *Cell Motil.*, 1985).

Could the authors provide a gallery of these observations as well as a quantification of the occurrence of protofilaments inside microtubules?

We now provide a gallery of selected events showing luminal filaments inside neuronal microtubules (Supplementary Fig S6d).

It is important to note that <10% of these events were observed *in vitro* (Supplementary Fig 2a). Quantifying the occurrence of these specific MT architectures in cells is challenging, however, because of the low signal-to-noise ratio in our tomograms (which is related to sample thickness) and the presence of numerous MIPs in the MT lumen that may obscure elongated luminal structures. We observed only a few of these events *in vivo* but may have underestimated their frequency. For these reasons, we cannot provide an accurate quantification of the occurrence of luminal protofilament in these conditions. We acknowledge this limitation in the main text (p.7): “We observed a few rare examples of singlet microtubule structures containing filamentous densities...” as well as in the Material and Methods section (p.13 & 14).

Could the authors try subtomogram averaging to strengthen their claim?

As suggested by the reviewer, we attempted subtomogram averaging on *in vivo* singlet microtubules containing luminal protofilaments. Unfortunately, the resulting models were of poor quality (as shown below, the corresponding mrc files are available at the following link: <https://filesender.renater.fr/?s=download&token=52443abf-5bbc-4bb2-81e7-953b7f12d99b>). For this reason, we chose not to include these models and prefer to remain cautious in our interpretation of the filamentous densities observed in neuronal microtubules.

Although the authors do not claim that MAP6d1 is needed for luminal protofilaments in vivo, it would be interesting to know if similar observations can be made in MAP6d1-KO cells.

We agree, but, as noted above, luminal protofilaments are already rare and difficult to detect by cryo-ET, even in wild-type neurons. Given these technical limitations, we believe that such an analysis would be unlikely to yield definitive conclusions regarding the role of MAP6d1 in the *in vivo* formation of luminal protofilaments.

In addition, the quality of visualization for in vivo tomograms would improve a lot if a denoising program is used such as IsoNet or CryoCare or DeepDeWedge.

While denoising increases the *contrast* of tomograms, facilitating, for example, the selection of globular MIPs, it also decreases the *resolution*. In our case, denoising made it difficult to distinguish the luminal filamentous density from the strong signal of the microtubule wall.

In addition, image processing must be better described. There are a lot of details missing in the image processing. Perhaps a diagram showing the processing workflow is better describe it.

We appreciate the suggestion to diagram the workflow. We have now included more detailed information on image processing in the Material and Methods section. We also provide the requested diagram in Supplementary Fig 7.

Is the search used filament properties to limit the search angles?

We used Eman2's filament option to select the microtubules and extract particles. In Eman2, the “—curve” option preserves the directional information of the filament and uses it for the orientation search of particles in the refinement step. We performed alignment and refinement of the subtomograms both with and without the “—curve” option. We obtained a slightly better reconstruction when this option was not applied.

It was puzzling to me that the maps provided by the authors have a very weird orientation. Normally, when you have a filament, the filament axis must be along the Z axis. That allows better comparison of the authors' structure with other structures.

Thank you for pointing this out—we have corrected the orientation of the refined maps of the SMT and DMT with luminal pf, ensuring that the filament axis aligns with the Z axis. We have updated the structures in the EM database and the corresponding mrc file can be downloaded with the following link: <https://filesender.renater.fr/?s=download&token=52443abf-5bbc-4bb2-81e7-953b7f12d99b>.

There are not clear how the singlets with different protofilament numbers are classified, how do the singlets with luminal pf are aligned, and any variability in the number of luminal pfs (from the authors' image, I could see that might be 1 or 2 pfs).

As indicated in the Material and Methods section and illustrated in the added workflow, we verified that all singlet microtubules contained two luminal protofilaments prior to 3D-refinement. This was done through 3D classification using 14-protofilament microtubules with either one or two luminal protofilaments as reference models.

Does the author do focus refinement on the luminal space, allowing better resolution of the luminal protofilament and more confident fitting?

We thank the reviewer for this excellent suggestion. We have now applied focused refinement around the two luminal protofilaments using a mask, and the resulting map is indeed much more convincing. This has significantly improved the model of the two luminal protofilaments. These results are now included in Figure 5c, and we also deposited the model in the EM database. The mrc file of the map is also available on the following link: <https://filesender.renater.fr/?s=download&token=52443abf-5bbc-4bb2-81e7-953b7f12d99b>.

Supplementary Fig. 1 shows that a mutant lacking the second Mn2 domain has a greatly reduced affinity for microtubules. What is the role of the first Mn2 domain? Would it make sense to create a mutant of the first Mn2 domain and study its effects on microtubule stability?

As suggested by the reviewer, we designed a mutant of MAP6d1 with seven alanine substitutions in the first Mn motif (named as MAP6d1-Mn1-7A). Our results show that this mutant has reduced affinity for MTs compared to MAP6d1, weakly stabilizes MTs, and fails to recruit tubulin or

induce MT doublets. In contrast to the MAP6d1-Mn2-7A mutant, MAP6d1-Mn1-7A retains some residual MT-stabilizing activity (decreased shrinkage rate, increased rescue frequency), suggesting that the Mn2 motif plays a dominant role in the MT-regulating activity of MAP6d1. These results are presented in Supplementary Figure 4 of the revised version.

I think Fig. 3 needs a more complete description along with more statistics in panel b to compare the mutants. If the conclusion is that the deltaMn mutant and the Mn-7A don't stabilise microtubules, then one would expect no significant difference between mutants and control MTs.

To avoid overcrowding the figure, we had initially indicated only the statistically significant differences in the legend. We have now added the non-significant differences as well. The two mutants, Δ Mn2 and Mn2-7A, show no significant difference from the control in their effects on shrinkage rate, catastrophe, or rescue events, although they slightly increase the growth rate. Therefore, we conclude that these mutants do not stabilize MTs.

MAP6d1 delatN stabilises microtubules and reduces catastrophe frequency while the WT seems to increase it. The same mutant also seems to increase the growth rate. Could these observations suggest an additional role for the N terminus to its requirement for microtubule pausing?

We are not sure we follow the reviewer's line of thought here. MAP6d1 induces pauses, a process associated with the progressive inhibition of polymerization and depolymerization rates. Accordingly, deletion of the N-terminal region of MAP6d1 significantly promotes the growth rate and therefore abolishes the pausing effect. Thus, it seems reasonable to link the N-terminal domain of MAP6d1 to MT growth control and pausing. We now mention in the text this stimulating effect of MAP6d1- Δ 2-35 on the growth rate (p.5).

Minor suggestions

Page 3: Cite Supplementary Fig. 1 before Fig. 2

Thank you; done.

Page 4: Cite Supplementary Fig. 2b before Supplementary Fig. 3.

Thank you; done.

Page 6: Ref 41 is for mitosis.

We removed this reference.

Page 6: ...adjacent to the putative seam (Fig 5c) -> Fig 5b should be cited

Corrected.

Page 8: KIF21B: ref 45 should be cited

Done.

Fig 1: panel a, seed should be moved to the left.

Done.

Fig 2c: very pixelated

This is because the lines have been drawn in Image J with one-pixel thick lines.

Fig 4: “stained for” instead of “stained against”

Thanks; corrected.

Fig 5: write “putative seam” instead of seam in Figure 5”

Done.

Fig S1: Remove extra WB, add MW next to ladder.

Done.

Fig S2a: catastrophe frequency, the bar for the mean is missing for 20nm.

Corrected.

Fig S4a legend: circles are the same colour – also true for other figure legends.

Corrected.

Reviewer #2 (Remarks to the Author)

Reviewer #3 (Remarks to the Author)

In this manuscript, Gopal and colleagues explore the functions of MAP6d1 protein, also called SL21 (STOP-like 21), member of the Stable Tubule Only Polypeptide (STOP) family, and the Stabiliser of AXOnemal microtubules (SAXO) family.

Using TIRF microscopy and cryo-EM of the in vitro reconstituted components, the authors show that MAP6d1 is a potent MT stabilizer which induces pauses of MT growth/shrinkage, and that it induces MT doublet formation based on its ability to concentrate soluble tubulin on the lattice of single MTs. They further show that these properties are dependent on the Mn2 and N-terminal domains of MAP6d1. CryoEM of microtubules in vitro allows authors to demonstrate the presence of protofilaments present inside single or doublet MTs, a phenomenon that was never observed before and that might change our view on microtubule biology. One possible role of this filament might be to determine the localisation of the B-tubule, an exciting hypothesis the authors discuss. Moreover, the authors also observe intra-luminal protofilaments in MTs in primary neurons, providing evidence for the presence of this phenomenon in vivo. The authors suggest that these filaments could confer stability to neuronal, long-lived MTs. Finally, they demonstrate that in primary mouse neurons, MAP6d1- localised to the proximal part of the cilium - is necessary for cilia length control.

The manuscript is clear and well-written, the experiments are thoroughly described, and figures are assembled with care and are easy to understand. Analyses are meticulous and well documented.

The scientific question is interesting and timely, as the research into MIPs is still in its early stages. Furthermore, the described effects of MAP6d1 on the formation of intra-luminal protofilaments both in vitro and in cultured neurons are fascinating and new and could add to our understanding of how neuronal microtubule stability is achieved.

We appreciate the reviewer's enthusiasm for the work and their insightful comments, which we have done our best to address.

I believe that a better effort could be made to better distinguish between what MAP6d1 can do in vitro, and what really is its function in vivo, which is why I have a few questions/remarks, that would be important to address before the manuscript can be accepted for publication.

1. I believe it is important to distinguish what a protein can do in an in vitro setting, and what it really does in cells. In this respect, how do the authors reconcile the conclusion that MAP6d1 is important for cilia development, as cilia seem to develop before MAP6d1 is even expressed in mice? Accordingly, in DIV4 neurons there is no difference between cilia in WT and MAP6d1KO neurons. Could it rather be that MAP6d1 in vivo stabilizes cilia, rather than contributes to their development/building? Please revise the discussion accordingly.

We hypothesize that MAP6d1 is involved in the development of primary cilia, as both its depletion and overexpression affect cilia length. The reviewer is correct that MAP6d1 is expressed postnatally, but neuronal primary cilia continue to elongate and mature for several weeks after birth (Arellano et al., *J. Comp. Neurol.* 2012). We clarify this at the beginning of the Discussion (p.7).

2. In the buffer perfusion experiments (Fig1d): is MAP6d1 present in the perfused buffer? If not, does it not get diluted out?

MAP6d1 is not present in the perfused buffer, as the goal of this experiment was to assess the stability of MTs assembled with MAP6d1 upon dilution. Perfusion of the buffer could wash away some MAP6d1 molecules bound to MTs, which might explain why some MTs slowly depolymerize after buffer perfusion while other remain in a paused state. In both cases, a stabilizing effect of MAP6d1 (either through a paused state or a slow shrinkage) is still observed, indicating that MAP6d1 binding to MTs is persistent.

3. Where do the percentages given while discussing Fig 3c/d come from (1%-40% and 9% - 57%)? I can't see those values in the graphs.

We thank the reviewer for catching that we'd inadvertently reported the wrong values. We have now corrected them as follows: 1% vs 45% for the doublet proportion, and 10% vs 61% for tubulin recruitment (MAP6d1- Δ 2-35 vs MAP6d1, Fig. 3d). We also identified an error in the transcription of the data about tubulin recruitment for MAP6d1 in Fig. 3d, where some recruitment and no-recruitment values were inadvertently mixed. This resulted in an incorrect reported mean of 50%, whereas the correct value is 61%. We became aware of this mistake during the preparation of the source data, and have corrected it accordingly. Importantly, this correction does not alter the final results or the statistical significance.

4. Fig 4: can the authors show by EM that MAP6d1 really colocalizes with the doublet MTs in the cilium?

We tried several strategies, but were unsuccessful. To access the high-resolution structure of neuronal cilia and identify MAP6d1, as previously performed on MT doublets from flagella and motile cilia (Lung et al., *Cell* 2023), we attempted to purify primary cilia from cultured primary hippocampal neurons, a procedure not previously documented in the literature. We adapted several protocols for cilia purification from cell lines, including mechanical shear (Mitchell, *Current Protocols in Cell Biology* 2013; Mohieldin et al., *Advanced Science* 2020), calcium shock (Raychowdhury et al., *The Journal of Biological Chemistry* 2005; Corkins et al., *PloS One* 2019; Scarinci et al., *Frontiers in Molecular Biosciences* 2023) and peel-off/slide-pull techniques (Huang et al., *American Journal of Physiology. Gastrointestinal and Liver Physiology* 2006). Unfortunately, these approaches proved unsuitable for neuronal cultures and the ciliary yield was negligible.

We next thought to localize MAPs and MIPs in ciliary axonemes using EM and immunolabelling, but the ciliary inner compartment and the MT lumen are not very accessible. We

attempted immunolabelling using an anti-GFP antibody in neurons expressing MAP6d1-GFP in cilia, but no specific signal was detected within cilia because of the compartment's poor accessibility.

These difficulties with EM and immunolabelling are what led us to use the PTMs as markers of doublet versus singlet MTs, as already performed in previous studies (see for example Tran et al., 2024, *Dev. Cell*). To access a better resolution, we finally performed 4X-expansion microscopy, which indicates that MAP6d1 clearly co-localizes with polyglutamylated tubulin, known to be enriched on doublet MTs, in the central region of the cilia, positioned between the two membranes that delineate the borders of the cilia. We have added these expansion microscopy experiments in the revised version of the manuscript (Fig. 4d).

Do you predict that in DIV9 MAP6d1-KO cilia there are no doublet MTs left, while after MAP6d1-Δ2-35 overexpression the doublet MTs go up to the tip of the cilium (Fig 4c)?

To answer the reviewer, we examined the distribution of polyglutamylated and tyrosinated tubulin, which are associated with doublet and singlet architectures, in DIV9 WT and MAP6d1-KO neurons (Supplementary Figure 5d) as well as in neurons overexpressing MAP6d1-Delta2-35 (Supplementary Figure 5e). Our results show a similar distribution of polyglutamylated and tyrosinated tubulin in all types of neurons, although the cilia have different lengths, suggesting that MAP6d1 expression does not affect the doublet / singlet microtubule ratio.

5. Fig 5: Did the authors ever observe intraluminal protofilaments when MAP6d1 is added to already-polymerised MTs?

We performed experiments using pre-polymerized GMPCPP-stabilized MTs to investigate whether MAP6d1 promotes tubulin recruitment onto these MTs (Figs. 2 and 3). Under these conditions, microtubules polymerize at both ends of the GMPCPP-MTs, as illustrated in Figs. 2e and 3c. However, since electron microscopy cannot distinguish between pre-existing and newly assembled MTs, it is not possible to determine whether luminal protofilaments are also present in the pre-polymerized MTs. As noted in the text (p.7), we did not observe any obvious lattice defects near the luminal protofilaments, suggesting that these structures likely form during MAP6d1-induced polymerisation rather than through diffusion via lattice breaks.

6. Fig 5: How were the radii of the B-tubules measured? What is the radius of the B-tubule in vivo? It would be interesting to compare this value to the values measured in vitro.

The procedure for measuring the radii of the B-tubules is now fully detailed in the Material and Methods section (p.14). The radius of the B tubule in *Chlamydomonas* flagellar doublet microtubule is 13.5 μm, which is similar to that of the B-tubule in *in vitro* doublet microtubules assembled in the presence of MAP6d1. This information has been added in the legend of Fig 2c and in the text p.4.

Textual changes:

1. Please add in the introduction the alternative names of MAP6d1.

Done.

2. The sentence p.2: MAP6, a microtubule-stabilising factor linked to psychiatric disorders^{24,25}, was the first neuronal MIP located in the microtubule lumen, where it generates highly stable microtubules that grow in a helicoidal pattern²⁶. implies that MAP6 induces the growth of helicoidal MTs inside the lumen of the MT lumen: please rephrase.

We rephrased the sentence as follows: “MAP6, a microtubule-stabilising factor linked to psychiatric disorders, was the first neuronal MIP identified within the microtubule lumen. Intraluminal MAP6 generates highly stable microtubules that grow in a helicoidal pattern.”

3. In the same paragraph the word "motif" is repeated 4 times.

We modified the passage as follows: "MAP6 belongs to the SAXO (Stabiliser of AXOnemal microtubules) family of proteins that contain a small helical motif in their microtubule-binding domains known as the Mn-motif. This structural element has recently been recognized as a universal microtubule luminal binding feature in axonemal MIPs, highlighting its potential significance in stabilizing both axonemal and neuronal microtubules."

4. Title of the first results chapter: *MAP6d1 stabilises microtubules by inducing pauses: pauses of what?*

We changed the title as follows: "MAP6d1 induces microtubule pauses"

5. Last paragraph p.5: *for ectopic expression, please add information about time of transfection and time of analysis. Shouldn't "ΔMn2-35" be "Δ2-35"?*

Thank you, we corrected this.

6. 2nd paragraph p.6: *"performed immunofluorescence co-staining" not "co-immunofluorescence staining".*

Thank you, we corrected this.

7. Last paragraph of p.6: *remove "of doublet MTs".*

Done.

8. 2nd paragraph of the discussion, 2nd sentence: *a definition of a MAP does not include to be able to "recruit tubulin on the MT lattice", just like the definition of a MIP is not to "form intraluminal protofilaments". Please rephrase.*

The dual binding of MAP6d1 to microtubules as both a MAP and a MIP is a model we proposed based on our experimental data. While we acknowledge that recruiting tubulin to the microtubule lattice is not the definition of a MAP, our study implies that MAP6d1 binds to the microtubule surface like a classic MAP. Similarly, the ability of MAP6d1 to assemble luminal protofilaments and its two universal luminal-targeting Mn motifs suggest it functions as a MIP. We have rephrased our statement to be more precise as follows (p.8):

"Our data support a model in which MAP6d1 behaves both as a MAP, in that it binds to the surface of the microtubule lattice, and as a MIP, in that it contains two universal lumen-targeting Mn-motifs. These features, respectively, are necessary for its ability to recruit tubulin to the MT lattice and form intraluminal filaments. We propose that this "dual binding" mode is key to MAP6d1's unique ability to assemble doublet microtubules by bridging the A-tubule to the B-tubule (Fig. 6). »

Reviewer #1

Upon reading the manuscript again, I have some minor points that the author can address quickly before publication.

In their in vitro dataset, does the author encounter the doublet microtubule with more than 1 B-tubule (or more than 1 B-tubule hook)? From their dataset, DMT is about 20%. Therefore, the chance of having two hooks should be 4% if the hooks form independently. So, if the authors see no more than 1 hook in their dataset, there is a likelihood that MAP6d1 can be very specific to the 'putative seam'. So, a calculation and a statement on this in the paper would be good.

We observed only microtubule doublets with one B-tubule hook in our in vitro dataset. We added the reviewer's comment in the discussion (p.8, line 13):

"MAP6d1 binding is sufficient to restrict B-tubule nucleation to a single A-tubule protofilament. MAP6d1 therefore appears to provide positional information to recruit tubulin at a specific site on the A-tubule lattice, likely related to its binding near the seam region".

Is the subtomogram refinement Gold Standard Refinement (divided into two independent half sets and refined)? The reason is that if you use Gold Standard Refinement, then the FSC should be calculated at 0.143 not 0.5. FSC=0.5 is used in Non-Gold Standard Refinement. If Non-Gold Standard Refinement is used, the FSC should be clearly stated as Non-Gold Standard FSC or Semi-Independent FSC.

There are many studies using FSC=0.5 calculated from two randomly halves of the dataset to estimate resolution (for example: Davies et al. PNAS 2012; Schmidt-Cernohorska et al. 2019 Science). We added the estimated resolution at FSC=0.143 in the Figure S2 as requested by Reviewer 1. However, we retained the information at FSC=0.5, like presented by Scheres et Chen, Nat Methods 2012 or discussed by Steven Ludkte (<https://blake.bcm.edu/emanwiki/EMAN2/FAQ/FSC>), developer of the Eman2 package that we used to refine our subtomograms. In our opinion, the resolution estimated at FSC=0.5 is also more representative of the resolution of our models.